# Show and Write: Entity-aware Article Generation with Image Information

## Abstract

Prior work for article generation has primarily focused on generating articles using a human-written prompt to provide topical context and metadata about the article. However, for many applications, such as generating news stories, these articles are also often paired with images and their captions or alt-text, which in turn are based on real-world events and may reference many named entities that are difficult to be correctly recognized and predicted by language models. To address this, we introduce ENtity-aware article Generation with Image iNformation, ENGIN, to incorporate an article's image information into language models. ENGIN represents articles that can be conditioned on metadata used by prior work and information such as captions and named entities extracted from images. Our key contribution is a novel Entity-aware mechanism to help our model recognize and predict the entity names in articles, improving article generation. We perform experiments on three public datasets, GoodNews, VisualNews, and WikiText. Quantitative results show that our approach improves generated article perplexity by 4-5 points over the base models. Qualitative results demonstrate the text generated by ENGIN is more consistent with embedded article images. We also perform article quality annotation experiments on the generated articles to validate that our model produces higher-quality articles. Finally, we investigate the effect ENGIN has on methods that automatically detect machine-generated articles.

## 1 Introduction

Automatically writing articles is a complex and challenging language generation task. Developing a reliable article generation method enables a wide range of applications such as story generation (Fan et al., 2018; Peng et al., 2018), automated journalism (Leppänen et al., 2017; Brown et al., 2020), defending against misinformation (Zellers et al., 2020; Tan et al., 2020), writing Wiki articles (Banerjee & Mitra, 2016; Merity et al., 2016), among other applications. In early work (Lake et al., 2017; Jia & Liang, 2017; Alcorn et al., 2019), language models were trained using domain-specific data. These specialized methods worked well for in-domain data, but did not generalize to out-of-distribution inputs. To address this, language generators finetune large-scale pretrained language models (Radford et al., 2018; 2019; Brown et al., 2020) on domain-specific data such as news (Zellers et al., 2020) and Wikipedia (Merity et al., 2016). These methods can generate articles with unconditional sampling given the first few sentences of articles (Radford et al., 2018; 2019) or with conditional sampling given metadata such as title and author (Zellers et al., 2020; Brown et al., 2020).

There are two important challenges not explored in prior work for article generation. First, they only model text (Brown et al., 2020; Zellers et al., 2020) (Figure 1(a)), ignoring images embedded in the articles that may provide additional insights. Second, these methods only implicitly model named entities that commonly appear in long articles like organizations, places, and dates to provide context (Radford et al., 2019; Brown et al., 2020). These named entities are critical to accurately modeling a long article, but it often is not known what named entities may appear at test time.

To address these challenges, we propose an **EN**tity-aware article **G**eneration framework with **I**mage i**N**formation (ENGIN), which leverages image information and a novel entity-aware mechanism for article generation. Named entities indicate important contextual information about the events related to the news report in Figure 1(b). However, in prior work (Radford et al., 2019; Brown et al., 2020; Zellers et al., 2020) named entities are modeled together with the other text, and the language

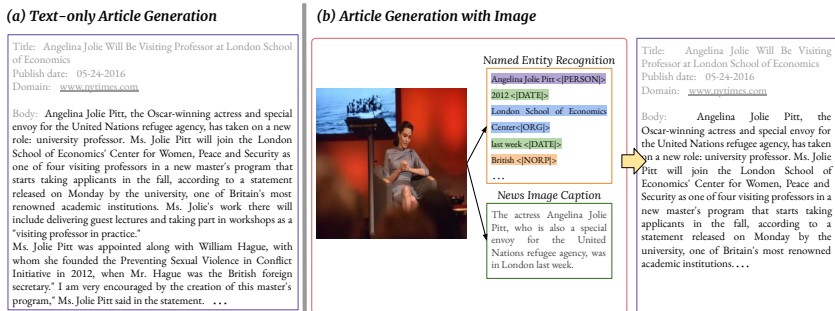

Figure 1: Prior work (Brown et al., 2020; Zellers et al., 2020), shown in (a), produces an article (black text) conditioned on article metadata (gray text), ignoring image information. This paper, shown in (b), also conditions on image information like extracted named entities, which may provide important context (*e.g.*, knowing the woman in the image is an actress), when generating articles.

model may find it difficult to distinguish entity names from the other text in articles. To solve this issue, we propose an entity-aware mechanism to help ENGIN recognize and predict named entities. Specifically, we insert special tokens after each entity name to indicate its entity category. ENGIN models the named entity with its entity category jointly. An additional benefit brought by our Entity-aware mechanism is the named-entity recognition (NER) ability, *i.e.*, our model not only recognizes and predicts the entity names but also predicts the entity category simultaneously.

Prior work has proposed entity-aware mechanisms for related tasks like news image captioning (*e.g.*, (Biten et al., 2019; Tran et al., 2020; Liu et al., 2020)). However, these methods do not generalize to article generation as they rely on having a lot of contextual information (an article) as well as a direct indication of what entity they need to generate a caption for (from the image). In contrast, when generating an article conditioned on a collection of images and captions, the model has to select when to use each entity in the metadata. In addition, some key entities may not be present in the metadata. For example, in Figure 1(a) the article mentioned Angelina Jolie Pitt is an Oscar winner, but this entity doesn't appear in the image or caption shown in Figure 1(b). In news image captioning, entities used in the caption almost always appear in the body of the article (Liu et al., 2020; Tan et al., 2020). Thus, as we will show, adapting entity-aware mechanisms used in prior work (*e.g.*, (Liu et al., 2020; Dong et al., 2021)) results in poor performance on our task.

While we show that providing a list of named entities for generating an article boosts performance, it does require a small overhead cost. To address this, we also show gains without manual input. Specifically, as shown in Figure 1(b), images and captions can contain important events or key figures associated with an article. Since large vision-language models have seen various named entities during training, we use CLIP (Radford et al., 2021) to automatically select a set of likely named entities from an image (see Section 3). Figure 2 presents the overall pipeline of ENGIN.

In summary, the contributions of this paper are:
- We propose an entity-aware language model called ENGIN for article generation. Compared to existing models focusing on text-only context (Zellers et al., 2020; Brown et al., 2020), ENGIN effectively leverages information from images and captions to generate high-quality articles.
- We propose an Entity-aware mechanism to help language models better recognize and predict named entities by also modeling entity categories, boosting performance.
- Experiments on GoodNews (Biten et al., 2019) and VisualNews (Liu et al., 2020) report 1.5B params ENGIN-XL boosts perplexity by 2.5pts over 6B param GPT-J (Wang & Komatsuzaki, 2021). We also show ENGIN generalize via zero-shot transfer to WikiText (Merity et al., 2016).
- We perform a user study to verify that ENGIN produces more realistic news compared to prior work (Radford et al., 2019; Zellers et al., 2020). This suggests that our model may help provide additional training data for learning more powerful machine-generated text detectors.

## 2  RELATED WORK

**Article Generation** in recent research produces text using large-scale pretrained transformer models, which can be divided into two categories: unconditional text generation (Radford et al., 2018;

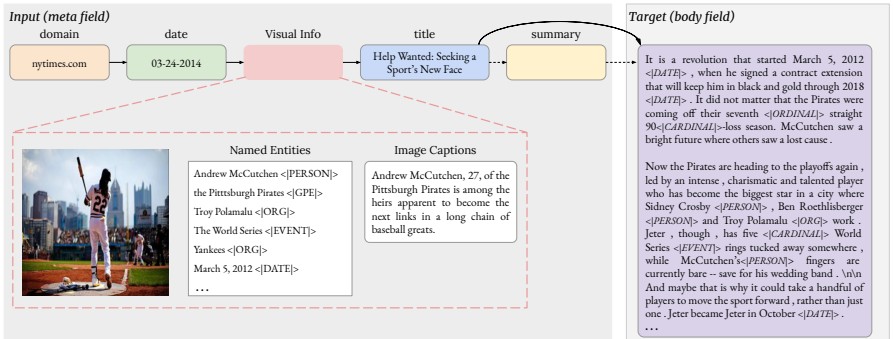

Figure 2: **ENGIN overview**. ENGIN generates articles conditioned on (1) meta field of articles including *domain*, *date*, *title*, *summary*, and so on (Section 3.1); and (2) vision field that consists of captions and named entities extracted from the embedded images (Section 3.2). Empty fields are skipped (like *summary* in this example) by using the start token of the next field. In addition, we propose an entity-aware mechanism (Section 3.3) to better represent named entities in articles.

2019) and conditional text generation (Brown et al., 2020; Zellers et al., 2020). Generating articles via unconditional samples has been shown to be less effective because the models may interpret the first sentence of articles as a tweet and start posting responses (Brown et al., 2020). To enable controllable generation, GPT3 (Brown et al., 2020) generates articles conditioned on titles, Grover (Zellers et al., 2020) decomposes news articles into separate parts and generates articles conditioned on the metadata like the author or organization. In this paper, we further explore the effect of visual information and named entities. Specifically, ENGIN produces articles conditioned on both the metadata like prior work and the visual information extracted from embedded images, with special care for explicitly extracting and modeling named entities.

**Article-based Image Captioning** such as news image captioning is designed to caption images based on articles and images. Ramisa et al. (2017) proposed an end-to-end framework that takes the concatenation of article and image features as input and outputs captions by an LSTM decoder. However, they often failed to predict named entities that were not seen during training. Thus, more recent work has included entity-aware mechanisms (Biten et al., 2019; Tran et al., 2020; Liu et al., 2020) to more accurately model these elements. In this paper, we effectively reverse the inputs and outputs of these papers, *i.e.*, we generate an article based on images and captions rather than generating captions based on images and articles. As discussed in the introduction, this shift breaks the assumptions used by entity-aware mechanisms in image captioning, so they do not generalize.

## 3 ENGIN: ENTITY-AWARE ARTICLE GENERATION WITH IMAGE INFORMATION

The goal of our task is to generate articles via conditional samples. In this paper, we propose ENGIN, which leverages image information and entity-aware mechanism to generate higher-quality articles. The input to ENGIN consists of both metadata and image information that is composed of named entities and captions (summarized in Figure 2). Section 3.1 introduces the article generation problem in detail. Section 3.2 discusses how we leverage image information. Section 3.3 presents our novel entity-aware mechanism. Finally, Section 3.4 summarizes the overall training strategy of ENGIN.

### 3.1 ARTICLE GENERATION

Given a set of documents $\{x_1, x_2, ..., x_n\}$ each consists of variable length sequences of symbols $\{s_1, s_2, ..., s_m\}$, the statistical language model of a text document $x$ can be represented by the product of the conditional probability of next symbol given all the previous ones (Bengio et al., 2003):

$$p(x) = \prod_{i=1}^{m} p(s_i | s_1, ..., s_{i-1}) \tag{1}$$

where each symbol $s_i$ is processed uniformly and the document $x$ is viewed as an unstructured *text* field (also referred as body field later). Language models based only on Eq. 1 produce articles via unconditional samples. As a result, these models cannot output controllable generation (Hu et al., 2017). For controllable generation, the language model can be formulated by the joint distribution of separate fields decomposed from the article $x$ (Zellers et al., 2020):

$$p(x) = p(\text{meta}, \text{body}) \tag{2}$$

where meta field is a data-dependent term consisting of a set of subfields. For instance, meta includes *date*, *title*, *summary* in GoodNews (Biten et al., 2019) and *domain*, *date*, *topic*, *title* in VisualNews (Liu et al., 2020). Thus, we model $x$ by:

$$p(x) = p(\text{body} \mid \text{meta})p(\text{meta}). \tag{3}$$

According to Eq. 3, we introduce special tokens <start-$\tau$> and <end-$\tau$> to indicate the boundaries of field $\tau$. The content of a target field $\tau$ is sampled from the model starting with <start-$\tau$> and ending with <end-$\tau$>.

## 3.2 Extracting Information from Images

One of the advantages of Engin over prior work (*e.g.*, (Radford et al., 2019; Zellers et al., 2020)) is that we can take advantage of the information provided by embedded images in articles. The current state-of-the-art that combines the image and language descriptions adopts the Transformer (Vaswani et al., 2017) architecture and models the text and image tokens as a single stream of data (Ramesh et al., 2021). However, we found using images directly is less effective in the article generation setting. Unlike typical image captioning systems, articles such as news or Wikipedia report events happening in the real world and tend not to describe objects in embedded images directly[1]. In addition, methods like (Ramesh et al., 2021) take 1024 tokens to encode the image. If we adopt this image encoding, the sequence length of our language models will be doubled and the model weights cannot be initialized from the standard pretrained language models. However, this is not to say the images contain no useful information. Indeed, as illustrated in Figure 1, many images contain information about named entities. Thus, as illustrated in Figure 2, we use named-entities extracted from images in our model, along with the ground truth image captions directly.

We provide information from images to our model via a combination of two fields, *caption* field and *named-entity* field. We use ground truth image captions directly as the content of *caption* field. Below we discuss two ways to build *named-entity* field.

**Oracle named-entities.** This approach assumes we are provided with all the named entities that would appear in articles. To accomplish this, we extract named entities from news articles using SpaCy (Honnibal & Montani, 2017), which is input directly to our model.

**CLIP-based NER.** Most existing methods identify a predefined list of named entities (Yadav & Bethard, 2019; Li et al., 2020). However, we note that CLIP (Radford et al., 2021) was trained on 400 million image-text pairs collected from the internet, many of which likely contained named entities. Thus, we use CLIP to build an open-ended visual NER framework. First, we construct the NER candidate list for each image by extracting named entities from the dataset articles using SpaCy (Honnibal & Montani, 2017). Then we use CLIP to predict the similarity between the article image(s) and the candidate entities. The top $k^2$ entities are then provided as input to our model.

## 3.3 Entity-aware Mechanism

As discussed in the Introduction, accurately generating named entities can help avoid inconsistencies between the images associated with an article and the article text itself. For example, in NBA news, an entity-aware model should be able to predict "Curry" given the previous word "Stephen" while

---

[1]For example, the language description of the image in Figure 2 is "a baseball player swing a bat at a ball" (by NIC (Vinyals et al., 2015)). However, the ground truth caption is "Andrew McCutchen, 27, of the Pittsburgh Pirates is among the heirs apparent to become the next links in a long chain of baseball greats" and the article is discussing Andrew McCutchen's contract and his future career.

[2]We set $k$ to 10 in this paper.

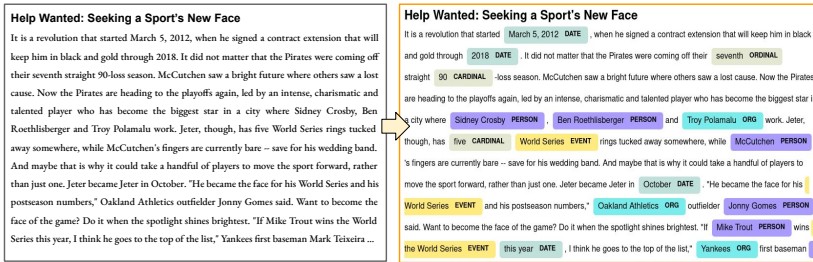

Figure 3: Example of the entity-aware mechanism in ENGIN. Each named entity is attached by its corresponding entity category. ENGIN models the entity name and category jointly to avoid the inconsistency between named entities of images and article text.

the traditional language models might fail. Existing methods model named entities uniformly with the other text, making the leverage of named entities less effective. To help our language model be aware of named entities, we insert the entity category predicted by SpaCy (Honnibal & Montani, 2017) after each entity name. We use special tokens as the indicator of these entity types. Then the entity name and its corresponding category are modeled jointly by ENGIN. We visualize our entity-aware mechanism by an example in Figure 3.

### 3.4 OVERALL LEARNING STRATEGY

**Architecture.** We build ENGIN using the same architecture of GPT2 (Radford et al., 2019). Following GROVER (Zellers et al., 2020), we propose three model sizes: (1) ENGIN-Base has 12 layers and 124 million parameters, on par with GPT2-124M and GROVER-Base; (2) ENGIN-Medium has 24 layers and 355 million parameters, on par with GPT2-355M and GROVER-Large; (3) ENGIN-XL has 48 layers and 1.5 billion parameters, on par with GPT2-1.5B and GROVER-Mega.

**Encoding Strategy.** Given the image information extracted from the embedded image, Equation 3 is re-formulated as:

$$p(x) = p(\text{ body } | \text{ meta }, \text{ vision })p(\text{ meta }, \text{ vision }) \qquad (4)$$

where vision field consists of *caption* field and *named-entity* field (from Section 3.2). To sample from Equation 4, we define a canonical order[3] among the fields (or subfields) of articles $\mathcal{F} : (f_1 < f_2 < ... < f_{|\mathcal{F}|})$ and model the articles left-to-right in the order using Equation 1: $s_1^{f_1}, s_2^{f_2}, ..., s_{|f_{|\mathcal{F}|}|}^{f_{|\mathcal{F}|}}$. If a specific field $f_i$ is missing, our model will automatically skip that field by introducing the start token of next field $f_{i+1}$. We illustrate such an example in Figure 2, where ENGIN starts generating the body field after *title* because the *summary* field is empty.

**Decoding Strategy.** Likelihood-maximization decoding strategies like greedy search or beam search work well in close-ended generation such as image captions, machine translation, or summarization. However, these methods suffer from the repetitive text problem in open-ended generations like dialog or story generation (Hashimoto et al., 2019; Holtzman et al., 2019). Sampling methods (Fan et al., 2018; Holtzman et al., 2019) are therefore proposed to introduce more randomness and surprise to text generation. In our work, we adopt the top-p sampling (nucleus sampling) method (Holtzman et al., 2019) as our decoding strategy for article generation.

## 4 EXPERIMENTS

### 4.1 DATASETS AND EXPERIMENT SETTINGS

**Datasets.** We evaluate ENGIN on three public datasets: GoodNews (Biten et al., 2019), Visual-News (Liu et al., 2020), and WikiText (Merity et al., 2016). The GoodNews dataset provides the

---

[3]We define canonical order in Goodnews (Biten et al., 2019) as: domain, date, named-entity, title, caption, summary, body; and Visualnews (Liu et al., 2020) as: domain, date, topic, named-entity, title, caption, body.

Table 1: Comparison of different generation methods and model sizes using perplexity (PPL) to measure performance on the GoodNews and VisualNews datasets. ClipNE denotes that we select CLIP-based named entities in *named-entity* field (described in Section 3.2), NE denotes that we apply oracle named entities in *named-entity* field. PPL is calculated only on the article body .

| Model Name | $n_{\text{params}}$ | $n_{\text{layers}}$ | $d_{\text{model}}$ | $n_{\text{heads}}$ | GoodNews PPL ↓ | VisualNews PPL ↓ |
|---|---|---|---|---|---|---|
| GPT2-124M (Radford et al., 2019) | 124M | 12 | 768 | 12 | 23.6 | 27.5 |
| GROVER-Base (Zellers et al., 2020) | 124M | 12 | 768 | 12 | 23.8 | 21.9 |
| GPT-Neo-125M (Gao et al., 2020) | 125M | 12 | 768 | 12 | 27.1 | 29.3 |
| GPT2-124M (Finetuned) | 124M | 12 | 768 | 12 | 17.3 | 18.3 |
| ENGIN-Base (ClipNE) | 124M | 12 | 768 | 12 | 14.8 | 16.1 |
| ENGIN-Base (NE) | 124M | 12 | 768 | 12 | **12.0** | **13.1** |
| GPT2-355M (Radford et al., 2019) | 355M | 24 | 1024 | 16 | 17.8 | 20.1 |
| GROVER-Large (Zellers et al., 2020) | 355M | 24 | 1024 | 16 | 18.5 | 16.4 |
| GPT-Neo-1.3B (Gao et al., 2020) | 1.3B | 24 | 2048 | 16 | 15.3 | 15.9 |
| GPT2-355M (Finetuned) | 355M | 24 | 1024 | 16 | 13.5 | 14.0 |
| ENGIN-Medium(ClipNE) | 355M | 24 | 1024 | 16 | 11.6 | 12.5 |
| ENGIN-Medium(NE) | 355M | 24 | 1024 | 16 | **9.5** | **10.2** |
| GPT2-1.5B (Radford et al., 2019) | 1.5B | 48 | 1600 | 25 | 13.9 | 15.7 |
| GROVER-Mega (Zellers et al., 2020) | 1.5B | 48 | 1600 | 25 | 14.5 | 12.6 |
| GPT-Neo-2.7B (Gao et al., 2020) | 2.7B | 32 | 2560 | 20 | 13.5 | 14.0 |
| GPT-J-6B (Wang & Komatsuzaki, 2021) | 6B | 28 | 4096 | 16 | 11.3 | 11.6 |
| GPT2-1.5B (Finetuned) | 1.5B | 48 | 1600 | 25 | 12.6 | 12.4 |
| ENGIN-XL(ClipNE) | 1.5B | 48 | 1600 | 25 | 10.8 | 11.1 |
| ENGIN-XL(NE) | 1.5B | 48 | 1600 | 25 | **8.7** | **9.0** |

URLs of news from New York Times ranging from 2010 to 2018. We were able to download 307,286 news articles from the provided URLs. The remaining articles are either broken links or non-English articles. Following the split ratios of (Biten et al., 2019), we randomly split 15,365 articles in validation set, 30,728 articles in test set, and the rest articles in training set. The VisualNews dataset contains news articles from four news sources: *Guardian*, *BBC*, *USA Today*, and *Washington Post*. We obtain 582,194 news articles in total after we removed broken links and articles without metadata. Similarly, we get a 491,796 training set, 28,932 validation set, and a 57,889 test set. The WikiText dataset contains 600 Wikipedia articles in training set, 60 articles in validation set, and 60 articles in test set. We also performed zero-shot article generation on the test set of WikiText.

**Metrics.** We adopt Perplexity (PPL) to quantitatively evaluate our language models. Perplexity is defined as the exponentiated average negative log-likelihood of a sequence. Given Equation 1, the perplexity of $x$ is calculated by:

$$\text{PPL}(x) = \exp\left\{-\frac{1}{m}\sum_{i=1}^{m}\log p(s_i|s_1,...,s_{i-1})\right\} \tag{5}$$

where $s_1,...,s_i$ are ground truth tokens in $x$ and $p(\cdot)$ is the probability predicted by the model.

## 4.2 LANGUAGE MODELING RESULTS

**Perplexity.** Table 1 presents sizes, architectures, and perplexity results of different models on GoodNews (Biten et al., 2019) and VisualNews (Liu et al., 2020). We see that ENGINs of all three model sizes significantly outperform the baselines. On the base size, our model ENGIN-Base(NE) improves PPL over the original GPT2-124M model by a factor of 2 (23.6→12.0, 27.5 → 13.1). We draw three major conclusions from Table 1. First, the data distribution still plays an important role. Finetuned GPT2s improve PPL over the original GPT2s. The improvements become less obvious with a greater model size (VisualNews: 27.5→18.3 of base size; 15.7→12.4 of XL size). Second, ENGIN noticeably improves the performance over finetuned GPTs (4-5 perplexity points on both datasets), which

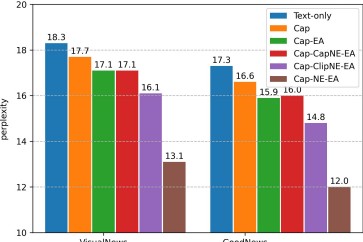

Figure 4: Ablation results of ENGIN-Base on GoodNews and VisualNews. Text-only denotes the model focuses only on text information, which is same to finetuned GPT2 model. Cap denotes *caption* field, EA denotes the Entity-aware mechanism. CapNE denotes named entities extracted from captions.

Table 2: Comparison of baselines adapted from close-related tasks using PPL. **(a)** leverages visual features directly extracted from images, **(b)** contrasts entity-aware mechanisms.

| Model Name | Good-News | Visual-News |
|---|---|---|
| **(a)** BU (Anderson et al., 2018) | 26.6 | 24.3 |
| **(b)** BDA (Post & Vilar, 2018) | 51.2 | - |
| InfoSurgeon (Fung et al., 2021) | 41.8 | - |
| InjType (Dong et al., 2021) | 18.2 | 19.0 |
| VNC (Liu et al., 2020) | 16.7 | 17.8 |
| ENGIN-Base(NE) | **12.0** | **13.1** |

demonstrates the effectiveness of our approach. Third, PPL improves with increased model sizes, indicating that a more powerful generator could be trained with even greater model sizes.

**Parameter Efficiency.** Table 1 shows that ENGIN can achieve a comparable performance with alternative models using much fewer parameters. For instance, ENGIN-Base(NE) only has 124M parameters but it outperforms the GPT-NEO-2.7B and achieves comparable performance with finetuned GPT2-1.5B (12.0 vs. 12.6 PPL on GoodNews, 13.1 vs. 12.4 PPL on VisualNews). ENGIN-Medium (NE) model with 355M parameters already outperforms all the baselines including GPT-J-6B.

**Additional baselines** adapted from closely-related tasks are presented in Table 2. BU[4] concatenates image embeddings extracted by the bottom-up features (Anderson et al., 2018) with meta information as input. InjType (Dong et al., 2021) applies a classification decoder to predict entity names from a list of candidate names. VNC (Liu et al., 2020) adds a named entity set to input as the entity-aware mechanism. BDA (Post & Vilar, 2018) and InfoSurgeon (Fung et al., 2021) apply copy mechanisms to sequence-to-sequence translation frameworks. In Table 2(a) we see that BU struggles to boost performance, likely due to the loose correlation between image content and their articles as discussed in Section 1. Table 2(b) demonstrates that the entity-aware mechanisms proposed for generating short text do not generalize to generating longer articles, where our approach obtains a 4-5 PPL improvement on both datasets. Besides, we find that BDA and InfoSurgeon do not perform well on article generation task. It may because the sequence-to-sequence translation frameworks of these models find it challenging to effectively leverage prior knowledge from pretrained language models (*e.g.*, (Radford et al., 2019)).

**Ablation Study.** Figure 4 shows ablations of ENGIN-Base to supplement our analysis. We observe that both the *caption* and *named-entity* fields boost performance, revealing that cues from news images can help produce higher-quality articles. Comparing using only captions (Cap) vs. combining them with our Entity-aware mechanism we get a minimum gain of 0.6 PPL, demonstrating its effectiveness. In addition, we observe that ClipNE outperforms CapNE, validating that CLIP-detected named entities are more effective than those extracted from captions.

**Article Quality User Study.** Following (Zellers et al., 2020; Kreps et al., 2020; Brown et al., 2020), we ask annotators to distinguish machine-generated articles from human-written articles. We randomly selected 50 news stories from GoodNews and VisualNews test sets (100 total). Given the metadata and news images, we generated news articles using three different language models: GROVER-Mega, GPT2-1.5B (finetuned), and ENGIN-XL. This results in a total of 200 articles per dataset. We recruited 200 Qualified Amazon Mechanical Turk (AMT) workers per news dataset. Each article was annotated 5 times by AMT workers, where each worker was presented with the

---

[4]InjType (Dong et al., 2021), BU (Anderson et al., 2018), and VNC (Liu et al., 2020) are adapted from close-ended paragraph generation, image captioning, and news image captioning, respectively. BDA (Post & Vilar, 2018) and InfoSurgeon (Fung et al., 2021) are adapted from sequence-to-sequence models.

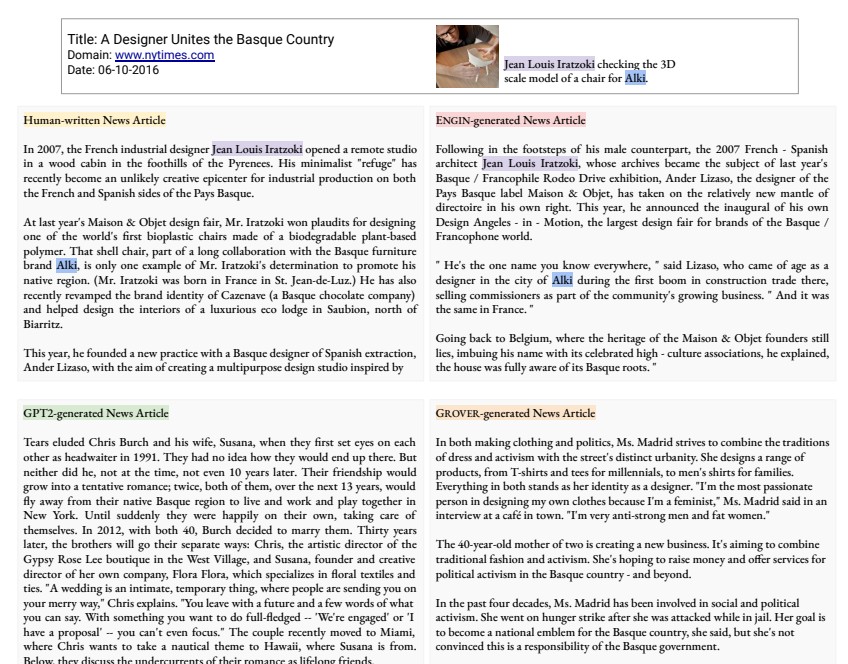

Figure 5: Qualitative examples comparing ENGIN-XL with GPT2-1.5B (Finetuned) and GROVER-Mega on GoodNews. We cut the articles to fit the figure size. The entity names from image information are highlighted in light purple (PERSON tag) and light blue (ORG tag) colors. We can see that the named entities in captions also appear in the human-written and ENGIN-generated articles. In contrast, the GPT2-generated and GROVER-generated articles do not contain the correct entity names corresponding to the image and caption.

article titles, images and captions, and was asked to indicate if the article was human or machine generated. If they thought the article was machine-generated, they were asked to indicate a reason for it following the same option format as Tan et al. (2020).

Table 3(a) reports annotation accuracy in identifying articles from VisualNews as machine or human-generated. We see ENGIN-XL is able to generate hard-to-detect news articles (we see a 2% boost over GPT2-1.5B, and a 5% gain over GROVER-Mega), validating the effectiveness of our approach.

**Machine Discriminator.** We also apply a machine discriminator to detect the generated articles. Specifically, we apply a RoBERTa (Liu et al., 2019) detector finetuned with the outputs of GPT2-1.5B (Radford et al., 2019) to detect generated articles. The maximum article length is cut to 512 to fit the model input size. For comparison, we use the same article set from our user study. The RoBERTa accuracy is shown in Table 3(b). We observe that the machine discriminator is much better at identifying the machine-generated news. We also see that articles produced by ENGIN-XL can be reliably detected by RoBERTa though it gets the lowest accuracy on human evaluation. This can be due to the fact that GROVER-Mega, GPT2-1.5B, and ENGIN-XL all share a similar underlying model architecture. Thus, they may contain enough similarities in the distributional features that are recognized by the machine discriminator.

**Zero-shot Article Generation** We perform zero-shot experiments on Wikipedia articles to demonstrate the generalization of ENGIN. Table 4 reports the perplexity points on WikiText. From the table, we see that our oracle named entities and entity-aware mechanism still can improve the performance over several baselines, even though the data distribution between Wikipedia and news is significantly different. For example, the GPT2 models finetuned on Visualnews get worse performance than the original GPT2 models on WikiText. However, our ENGIN models get comparable or better results than the original GPT2 across different model sizes.

**Qualitative Results.** We provide a qualitative comparison of GoodNews articles in Figure 5. Consistent with our annotation experiment, we compare the human-written article with three machine-generated articles. From the results, we can see that ENGIN-XL model can effectively produce

Table 3: **(a)** reports the performance of AMT workers at correctly identifying an article as human or machine generated over different generation methods while **(b)** uses OpenAI's machine generated text detector based on RoBERTa (Liu et al., 2019) to perform the same task.

| (a) | **Human-based detector** | | (b) | **RoBERTa detector** (Liu et al., 2019) | |
|---|---|---|---|---|---|
| | GROVER-Mega | 72.8% | | GROVER-Mega | 90% |
| | GPT2-1.5B (Finetuned) | 69.6% | | GPT2-1.5B (Finetuned) | 84% |
| | ENGIN-XL | 67.6% | | ENGIN-XL | 84% |

Table 4: Zero-shot results on WikiText (Merity et al., 2016). Both finetuned GPT2 and ENGIN models are finetuned based on VisualNews (Liu et al., 2020). The maximum length of Wikipedia articles is set to 1024.

| Method | $n_{\text{params}}$ | PPL↓ | $n_{\text{params}}$ | PPL↓ | $n_{\text{params}}$ | PPL↓ |
|---|---|---|---|---|---|---|
| GPT2 (Radford et al., 2019) | 124M | 26.1 | 355M | 19.1 | 1.5B | 14.8 |
| GPT-Neo (Gao et al., 2020) | 125M | 24.9 | 1.3B | **13.1** | 2.7B | 11.5 |
| GPT-J-6B (Wang & Komatsuzaki, 2021) | - | - | - | - | 6B | **9.0** |
| GPT2 (Finetuned) | 124M | 33.8 | 355M | 25.2 | 1.5B | 25.9 |
| ENGIN-Base (NE) | 124M | **20.7** | 355M | 15.4 | 1.5B | 16.3 |

articles with the named entities learned from image information. In contrast, finetuned GPT2-1.5B and GROVER-Mega failed to generate correct named entities in articles. For example, both ENGIN-generated article and the human-written article mentioned "Hean Louis Iratzoki" and "Alki", which are appeared in the caption. In contrast, articles generated by GPT2 or GROVER are discussing some other entities such as "Chris Burch" and "Ms. Madrid."

### 4.3 DISCUSSION

**Defending against machine-generated misinformation.** In our paper, we mainly investigate modeling machine-generated articles, which can be used directly for generation, but also can provide strong language features to support applications like article retrieval. However, actors can also use the same technology to generate articles with misinformation such as fake news by modifying information of specific fields to realize two purposes: monetization (ad revenue through clicks) or propaganda (communicating targeted information) (Zellers et al., 2020). Therefore, the development of a better article generator can not only help humans write high-quality articles but also potentially help train a more powerful discriminator. When comparing the results of the RoBERTa detector for GPT2-1.5B and ENGIN-XL in Table 3, we find that only 25% of the articles that were predicted as human written came from the same generation prompts. Thus, the two methods can provide different views given the same prompt, which can provide additional information for training an even more powerful machine generated text detector. We note that our contributions are largely architecture agnostic, so they could be provided as input to RNN-based generators, which may provide a larger distribution shift in the generated articles that may fool a discriminator trained only on Transformer-based outputs.

### 5 CONCLUSION

In this paper, we proposed an entity-aware article generation method called ENGIN to address two factors that are unexplored by prior work: image information and named entities. Concretely, ENGIN produces articles conditioned on both metadata and visual information extracted from embedded images in articles. Moreover, we introduced an entity-aware mechanism to help ENGIN recognize and predict named entities more effectively. ENGIN outperforms current popular language models using much fewer parameters in quantitative and qualitative experiments on GoodNews and VisualNews. For example, ENGIN-XL outperforms GPT-J by roughly 2.5 perplexity points using only a quarter parameters of GPT-J. The noticeable improvements demonstrate that ENGIN can generate articles more accurately and efficiently. The article quality annotation experiment further validates that ENGIN-generated articles have higher quality compared to existing methods.

## 6 REPRODUCIBILITY STATEMENT

We implemented our models mainly based on Pytorch (Paszke et al., 2019) and Transformer (Wolf et al., 2020) libraries. The maximum sequence length of language models is set to 1024. For ENGIN-Base and ENGIN-Medium, we used a batch size of 8 and a maximum learning rate of $1 \times 10^{-4}$. For ENGIN-XL, we used a batch size of 4 to fit the GPU memory. Correspondingly, the maximum learning rate is set to $2^{0.5} \times 10^{-4}$. We trained our models around 3 epochs with 0.06 epoch for linear warm-up on both datasets. We parallelized ENGIN-XL on 4 NVIDIA RTX-A6000s and ENGIN-Medium on 2 NVIDIA RTX-A6000s. The longest training time, ENGIN-XL on VisualNews, takes approximately two weeks on our system. We will also release our code to ensure reproducibility after the paper is accepted.

## 7 ETHICS STATEMENT

ENGIN is a model for article generation. It can either help automated journalism or defending against machine-generation articles. However, there is no perfect system which can generate 100% accurate articles. Therefore, it is critical for practitioners to check the fact mentioned in articles and avoid the misinformation brought by failure generation cases. Additionally, someone could use our approach to generate misinformation. However, ENGIN applies the network structure that is same to GPT2, which means the discriminators trained for GPT2 articles (*e.g.*, RoBERTa detector (Liu et al., 2019)) are also effective to discriminate ENGIN-generated articles. Our paper helps to highlight the need for building tools like the RoBERTa detector.

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

# A  ADDITIONAL EXPERIMENT RESULTS

## A.1  ARTICLE GENERATION

Additional qualitative results on article generation are provided in Figure 7, 8, 9, 10, 11, 12, 13 as the supplement of our main paper, demonstrating that ENGIN is able to generate articles given different news sources.

## A.2  ARTICLE QUALITY ANNOTATION

Table 5 reports human accuracy in identifying articles as machine or human-generated to supplement the main paper. We see that ENGIN-XL is able to generate more realistic-looking news articles on Goodnews, consistent with the conclusion in our main paper.

Table 5: The performance of AMT workers at correctly identifying an article as human or machine generated on GoodNews.

| Human-based detector | |
|---|---|
| GROVER-Mega | 76.4% |
| GPT2-1.5B (Finetuned) | 73.6% |
| ENGIN-XL | **70.4%** |

Table 6: Comparison to baselines with copy mechanism on GoodNews.

| Method | PPL $\downarrow$ |
|---|---|
| BDA (Post & Vilar, 2018) | 51.2 |
| InfoSurgeon (Fung et al., 2021) | 41.8 |
| ENGIN-Base | 12.0 |

### A.3 COMPARISON TO BASELINES WITH COPY MECHANISM

In our paper, we mainly explore the entity-aware mechanism in language modeling. Another way to include named entities in articles is applying copy mechanism (Post & Vilar, 2018) to the decoding process. In Table 6, we compare our method with BDA (Post & Vilar, 2018) and InfoSurgeon (Fung et al., 2021). We reproduce both baselines based on SOCKEYE (Hieber et al., 2017). In Table 6, We see that our model outperforms these baselines(51.2 vs. 41.8 vs. 12.0). There are two major reason that may drop the performance of these models. First, since BDA and InfoSurgeon are built based on sequence-to-sequence framework, these methods may not effectively leverage the prior knowledge from pretrained generative language models such as GPT2 (Radford et al., 2019). Second, though named entities are enforced to appear in the output, the models may not put all named entities in correct positions.

### A.4 ABLATION STUDY ON TOP-k NAMED ENTITIES

We provide the ablation study on top-k named entities in Table 7. We see that the model achieves the best performance when $k$ is set to 15. When $k$ is greater than 10, the improvement is limited.

### A.5 PARAMETER EFFICIENCY

Figure 6 plots the perplexity of language models on GoodNews and VisualNews as a function of the number of parameters, further validating that ENGIN models get comparable or better results to alternative methods using far fewer parameters.

### A.6 ARTICLE QUALITY ANNOTATION TEMPLATES

Following (Tan et al., 2020), annotators are asked to indicate a reason for whether the articles are human-written or machine-manipulated. We provide a view of the AMT worker interface in Figure 14.

### A.7 LANGUAGE FEATURES FOR ARTICLE-TO-IMAGE RETRIEVAL

To validate that the language features of our model can also be applied to downstream tasks, we provide additional experiments on article-to-image retrieval in Table 8. We train a linear probe between features computing using ENGIN-base and Finetuned GPT2 from Table 1 of our paper and image features. Specifically, image features are extracted by CLIP and language features are extracted from the last hidden layer of language models. Then, given an article, the goal is to retrieve the ground truth image from a set of 1K images. We see that ENGIN outperforms both the CLIP and MIL-NCE baselines used in Tan et al., 2022, as well as our own linear probe over finetuned GPT2. This helps demonstrate the ability of our approach to generalize to other downstream tasks.

Table 7: Ablation study on the number of named entities detected by CLIP (GoodNews).

| top-k named entities | 5 | 10 | 15 | 20 |
|---|---|---|---|---|
| PPL ↓ | | 15.5 | 14.8 | 14.5 | 14.6 |

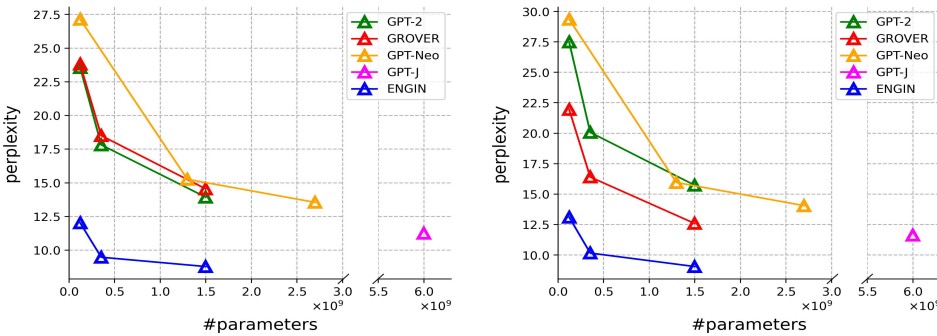

Figure 6: Comparison of the perplexity of different language models on GoodNews (left) and VisualNews (right) as a function of learned parameters.

Table 8: Article-to-image retrieval on GoodNews. We follow the experiment settings in NewsStories (Tan et al., 2022) for evaluation.

| Method | R@1 | R@5 | R@10 |
|---|---|---|---|
| Pretrained CLIP (Radford et al., 2021) | 18.43 | 36.59 | 46.92 |
| Single Image (Radford et al., 2021) | 17.14 | 33.77 | 43.56 |
| MIL-NCE (Miech et al., 2020) | 15.50 | 28.96 | 37.60 |
| Finetuned GPT2 | 16.08 | 31.99 | 41.45 |
| ENGIN | 18.24 | 35.62 | 45.36 |

## B LIMITATIONS AND FUTURE WORK

We discuss potential improvements to our work in this section. First, though our method can effectively predict the correct entity names in articles, their corresponding entity categories might be mistakenly predicted. For example, in Figure 5, the brand name "Alki" is recognized as a city name by ENGIN. Therefore, a more accurate entity-aware mechanism could be developed in future work. Second, the image information can be further explored. In this paper, we mainly investigate the captions and named entities of news images. However, other information such as the locations of images within articles may also prove useful for article generation. In addition, our current methods detect named entities from images considering each entity independently using a text-image matching framework. However, since the relationships between entities also affect the probability that entities appear in images, the incorporation of entity relationships can also be considered to further improve the entity detection module.

Title: Daniel Agger leaves Liverpool to return to former club Brondby
Domain: www.theguardian.com
Date: 08-30-2014

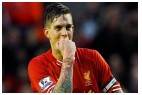 Daniel Agger has rejoined Brondby for an undisclosed fee.

**Human-written News Article**

Liverpool have announced the sale of the defender Daniel Agger to the Danish club Brondby for an undisclosed fee. The 29-year-old Denmark international joined the Anfield club from Brondby in 2006 and went on to make more than 200 appearances for the club but had fallen down the pecking order under Brendan Rodgers.

"Liverpool have been such a big part of my life and my family's lives for so long, that leaving is extremely difficult," Agger said. "The opportunity arose for me to return home to Brondby and at this stage of my career it felt like the right decision for me. "I wouldn't leave here to go anywhere else and that has been proven by my actions in recent seasons - I have turned down many offers to move to other Premier League and European clubs. "I would like to thank the Liverpool supporters for the incredible backing they have given me in my time here and the warmth and generosity of spirit displayed to my family. "This has been our home and somewhere we have been very happy and settled. However, more recently I have started to feel the effects of playing regularly in a league that is as physically demanding as the Premier League is, and I would not want to stay if I did not think that I was able to meet that challenge week in, week out."

**ENGIN-generated News Article**

The Liverpool defender Daniel Agger has left Anfield to return to his previous club Brondby. The Danish international midfielder was reported to have agreed personal terms with the club. The 29-year-old left Liverpool in 2006 following their relegation from the Premier League and has featured for the national team in Denmark's recent seasons.

The move to Brondby was said to be a " cash - plus " deal. " We're very pleased Daniel Agger has chosen to continue his development at Brondby, " read a statement on the club website. " He's a young man who has shown great potential in the Premier League, and we look forward to working with him again. " Brondby, who are in the Danish Superliga, have a squad of more than 200 players. The Liverpool manager Brendan Rodgers said : " As a footballer you dream of playing in the Premier League and I'm very pleased that Daniel Agger has chosen to come back to Brondby. " He's a good defender and I'm sure he will work very hard to prove to the club that he is a good player. " We like to think of him as a very important player for his nationality. But he has a great character. You might think he's a bit of a character but you 'll see he's as strong a character as any of the other players here. " Agger said : " I am extremely proud to play for Brondby. " I will always consider it as one of the happiest days of my career so far. It has been a great and very memorable time, and the supporters have been fantastic. My future lies with the club, and hopefully, it will be here for a long time. " I also want to thank the Liverpool fans for their support. Without them I wouldn't have been able to carry on fighting for this shirt, so thank you! "

Figure 7: Example of human-written and ENGIN-generated news articles from *Guardian*.

Title: State of Britain's wildlife 'increasingly fragile'
Domain: www.bbc.com
Date: 12-08-2015

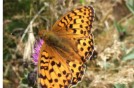 Dark green fritillary butterfly

**Human-written News Article**

Britain's wildlife is in an increasingly fragile state, with animals carrying out vital jobs for farmers being lost more rapidly than others, say scientists.Insects that pollinate crops or fight pests are at risk of disappearing, putting food production in jeopardy, according to the team.The research brings together millions of wildlife records spanning 40 years.It suggests conservation efforts should focus on certain areas, they add.Dr Tom Oliver of the University of Reading, who led the research, said it was the biggest and most comprehensive report ever assembled for any country in the world."By standardising records from an army of amateur biologists across the country, we have amassed an impressive array of data, giving us our most complete picture yet of the state of Britain's wildlife," he told BBC News. "The picture that emerges is of an increasingly fragile system, particularly in species that do vital jobs for humans. "Unless efforts are made to reverse some of these declines, we face a future where we will be less confident that we can effectively grow our food."Benefits from nature.

The researchers looked at records of the changing fortunes of more than 4,000 types of plant and animal living in England, Wales and Scotland between 1970 and 2009.Species such as bees, dragonflies, grasshoppers, ladybirds, plants and mosses were put into groups based on the role they play in nature.Groups providing pollination and pest control benefits had undergone declines while those involved in functions such as decay or mopping up carbon emissions were more stable.Plants and animals regarded as of cultural importance to humans, such as birds, butterflies and hedgehogs, also fared badly.Prof James Bullock, of the Centre for Ecology and Hydrology in Oxfordshire, a co-researcher on the study, said conservation efforts should focus on the areas where there was strong evidence of declining fortunes."Conservation actions, such as wildlife friendly farming, can avoid the loss of biodiversity and the resulting erosion of the pollination, pest control and other benefits we derive from nature," he said.And Dr Oliver said there were a number of measures that individuals could take to help to improve the future for wildlife in the British Isles.They include: Making gardens wildlife friendly Purchasing food grown in a sustainable way to wildlife Helping to collect data on wildlife. The study is published in the journal Nature Communications.

**ENGIN-generated News Article**

The state of Britain's wildlife has become increasingly fragile due to loss of habitat and changes in the climate, a study has suggested. Mass bird and butterfly losses have increased along with climate change, scientists found, which means many species are not as well equipped to cope. And while changes in food distribution, such as floods, may cause short - term problems, they can make long - term problems worse. Drought and flood events could put millions of birds and insects at risk, the scientists concluded. Monitoring of more than 4,000 species of birds, butterflies and moths across England, Wales and Scotland for 40 years concluded that the UK's biodiversity was still very good - only smaller than it was in 1975.But the biological communities in which they live were now less diverse than when these earlier surveys took place, which meant their ability to recover was less certain, they added. "We have lost several species of already endangered species, " explained lead researcher Dr Tom Oliver, a naturalist at the University of Reading. "And we are moving into many of these habitats that are becoming drier and thus less suitable for them. "The results of the study were " far more alarming " than expected. "The situation looks pretty dire, " said Dr James Bullock from the Centre for Ecology and Hydrology at Oxfordshire, who was not involved with the research. He told BBC News that Britain's wildlife was already extremely stressed. " Our natural environment is suffering from climate change, which is melting the ice sheets and expanding the oceans, and from human - generated pollution, so we are going to have a much more polluted environment in the future. " And the species we are losing are particularly important because when you lose species, you do so in numbers, they cease to exist in numbers. " The loss of a single species in Britain alone has the potential to affect millions of other organisms. "The impact of the findings is likely to be felt most in many communities across the British Isles. "We have lost a large number of species, " Dr Oliver added. " It's going to have a big impact on the ecology of Britain. " He explained that the risk of long - term damage was unlikely to be taken lightly : " We are more concerned about short - term impacts. "The researchers believe that much work needs to be done to understand how, in time, this situation might change. Research in Nature Communications suggests that climate change means there has been a sharp decline in the number of butterfly species spotted in northern England - down by a third between 1970 and 2009.Species such as the large blue have struggled for many generations as the cold weather has made it too cold to overwinter in the UK.Follow Helen on Twitter.

Figure 8: Example of human-written and ENGIN-generated news articles from *BBC*.

Title: Robberies plague Tenleytown area over the past few weeks
Domain: www.washingtonpost.com
Date: 05-23-2016

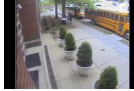 Video shows suspects robbing attacking
Georgetown Day student

**Human-written News Article**

A student at Georgetown Day School was robbed last month in the same area of the District where a student at another school was stabbed during a robbery attempt, and near where two other robberies also occurred last month.The student at Georgetown Day was robbed about 1:30 p.m. April 27 in the 4200 block of Davenport Street , near the private school. Three robbers approached, and began beating the victim, police said, although they did not identify him as a student.They went through his pants pocket and took his glasses, wallet and phone before fleeing on foot, police said.A spokeswoman for Georgetown Day issued a statement saying that a student there was robbed along 42nd Street in Tenleytown on April 27, and the school provided immediate medical and security support.The statement said the school has held an assembly devoted to neighborhood safety training and it has adjusted the patrols of school security officers.That robbery occurred two days after an April 25 incident in which authorities said a student at Woodrow Wilson High School was stabbed in the leg during an attempted robbery near the school.In the Wilson incident, the robber was described as a man in his 30s. No description was provided of the robbers on Davenport Street NW. Police released video Monday of three people described as "persons of interest."The video showed the victim and the robbers approaching each other on a sidewalk, with school buses lined up at a curb. Suddenly the student is knocked down and one or two of the robbers appear to go through his pockets.But the video does not appear to make possible a detailed description of the robbers.The stabbing comes nearly a week after two robberies, one of them involving a student victim, were reported on streets near Wilson and near Georgetown Day. Both occurred within about 15 minutes of each other April 19. In each instance, three people, one of whom who was armed with a knife, approached female victims and demanded their cellphones and debit cards. A juvenile male has been arrested in those cases, police said.

**ENGIN-generated News Article**

A group that robbed Georgetown Day School of valuables including cash last month and a purse while it was locked up in a dormitory has also robbed a spate of Georgetown Day students, police said. The initial thefts occurred April 27 and April 27. Two men entered the school, stealing the wallets and purses of two female students, police said. A total of four men were released from custody. The school said the total was $ 170.The last robbery occurred about 1:30 p.m. April 27 near the store at 4200 Davenport Street. Three male suspects fled the school on foot, and a local resident engaged in an effort to help the robbers, police said. It took about 15 minutes before the local man was able to locate them. The three defendants - one in his 20s, one in his late 20s and one in his late teens - fled along Davenport Street NW and into a residential neighborhood, police said. That resident was identified as a friend of the robbers and assisted in helping apprehend them. The robbery at Georgetown Day came two days after a robbery at Woodrow Wilson High School in Tenleytown. That April 25 robbery took place near 42nd Street and Davenport Street NW, a major thoroughfare in Tenleytown. According to police, two men in their 20s robbed a woman in her 30s. The robbers fled with a purse containing cash that was in a pink leather wallet. On Monday, the Wilson robbery went unsolved for nearly a week. Staff at the school said no incidents had been reported. "We do have a staff member watching the campus. You go through the normal procedures of setting up alarms, " said a Tenleytown police spokeswoman. " We were n't able to get that information from the yet. The person you are talking to is someone within security. You have to keep an eye on him. "Correction : This story has been updated to reflect that the Davenport Street robbery occurred April 19, not April 27.

Figure 9: Example of human-written and ENGIN-generated news articles from *Washington Post*.

Title: Courts again rule against streaming upstart Aereo
Domain: www.usatoday.com
Date: 08-21-2014

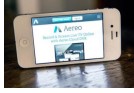 In this photo illustration Aereocom a Web service that provides television shows online is shown on an iPhone 4S on April 22 2014 in New York City

**Human-written News Article**

A federal court will not hear Aereo's argument that the streaming media company should be allowed to operate similarly to a cable company.The Second Circuit Court of Appeals ruled Thursday that Aereo will have to take its argument to the district court, according to court documents first reported by The Washington Post.This decision continues broadcasters' winning streak against the upstart service. Launched two years ago, Aereo delivered over-the-air channels via online streaming to computers and tablets for about $8 a month and had brought the service to 11 cities.The Supreme Court ruled two months ago that Aereo was violating copyright law by not paying the networks for transmitting their signals.After that, the streaming media company voluntarily shut down and told the U.S. District Court that it wanted to be treated as the equivalent of a cable TV system and apply for a compulsory license to stream live TV broadcasts.Subsequently, the Copyright Office's general counsel said that the office did not think that Aereo would qualify, but fell short of denying Aereo's filing because the issue remained alive in the courts.Follow Mike Snider on Twitter: @MikeSnider

**ENGIN-generated News Article**

The high court delivered yet another blow to Aereo in a pair of unanimous court opinions Thursday declaring the popular streaming service illegally alters the contents of customers'cable subscriptions and is guilty of violating copyright law. The Second Circuit Court of Appeals, which first ruled against Aereo two years ago, urged the justices to reject its claim that it is simply a device for consumers to access live TV without paying for it. Aereo lets consumers use a digital antenna to receive live broadcasts online and streams those broadcasts as they are being transmitted over the Internet, just like cable TV.Through those transmissions the consumer also becomes a subscriber to the over - the - air service. But even if they keep Aereo's service on, only a portion of the amount paid for cable service remains. For example, for $ 16.95 per month Aereo customers receive nearly all the TV shows they request, regardless of what shows they watch. For about $8 each the consumer loses the signals they might already have over the Internet. Thus the company is violating copyright law by redistributing the service users already pay for without paying for the service they are receiving from cable operators. "The Court finds that Aereo's business model violates this section of the Copyright Act, " 11 judges wrote in the majority opinion. The Washington Post, which first reported on the decision, called it " a blow to digital media and to consumers. ""This unanimous decision means a lot, " Aereo chief executive Mike Snider said in a statement. " Consumers no longer have to pay to receive the traditional way they watch TV. They now have the option to pay only for a service that delivers those channels, all for free. "Aereo has petitioned the court to change the district court's ruling to the full appeals court, which is likely to do so. The Supreme Court upheld the same ruling from two months ago. Aereo said it will appeal to the U.S. District Court for the District of Columbia. Experts tell the Post that its previous case shows that the Copyright Office's determination to give consumers access to streaming services without paying retailed companies is likely to be reversed by the appeals court. Follow @jswartz on Twitter

Figure 10: Example of human-written and ENGIN-generated news articles from *USA Today*.

Title: James F. Holland, Trailblazing Cancer Researcher, Dies at 92
Domain: www.nytimes.com
Date: 03-27-2018

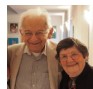 Dr. Holland with his wife, Dr. Jimmie Holland, who pioneered the field of psycho-oncology. While she treated the emotional distress of cancer patients, he focused on discovering universal remedies. She died in December.

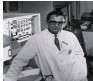 Dr. James F. Holland in an undated photograph. In the 1950s, he and his colleagues were considered research cowboys by some colleagues, he said.

**Human-written News Article**

James F. Holland, a founding father of chemotherapy who helped pioneer a lifesaving drug treatment for pediatric leukemia patients, died on Thursday at his home in Scarsdale, N.Y. He was 92.

The cause was respiratory failure, his son Steven said.

Dr. Holland and his medical collaborators in the 1950s were scorned by some colleagues as research "cowboys," as he put it, because they audaciously tackled what was considered an incurable disease and tried to treat it with several chemicals simultaneously, instead of sequentially.

When their research began, fewer than one in three children who were found to have acute lymphoblastic leukemia lived as long as a year. Today, as many as nine in 10 survive the disease, in which the bone marrow makes too many immature white blood cells.

Another clinical trial that Dr. Holland directed in the 1970s, to contain acute myeloid leukemia, a somewhat similar malfunction in blood cell production, resulted in what became the standard treatment.

**ENGIN-generated News Article**

James F. Holland, an innovative oncologist whose scientific insights made him, as he once said, a " Google " for cancer research, died on Thursday at his home in Scarsdale, N.Y. He was 92.

The cause was a widespread myeloid leukemia subtype, a cancer of the blood cells in bone marrow that occurs in fewer than one in three patients.

Mr. Holland was one of the few oncologists to propose, in the 1950s, the chemical treatment known as stereosporin -- hence the name. He was also among the first to suggest ( though not actually carry out ) what became known as precancer therapy : turning over nonmedical patients to their regular doctors, while supplementing their therapies with medications to kill cancer cells. The mere suggestion of precancer therapy, it seemed to some, would have rendered it toxic.

At the time, though, Mr. Holland thought it should not. He and his colleagues had been working for as long as a year on drugs to prevent acute lymphoblastic leukemia in patients with advanced disease, in which cells on the surface of the blood can proliferate out of control after being overexposed by chemotherapy. Doctors usually did this only after the patient's health deteriorated. Today, many hospitals will not perform precancer therapy, either.

But in the 1970s, he and his colleagues were regarded as " research cowboys " by their colleagues. Known for treating as many as nine of 10 cancer cases, they had been derided even by their own specialty, nuclear medicine, for treating these patients when " the science does not justify it, " Dr. Steven Holland, his brother and his only sibling, said.

Figure 11: Example of human-written and ENGIN-generated news articles from *New York Times*.

**Robert Boulter**

| Human-written Wikipedia | ENGIN-generated Wikipedia |
|---|---|
| Robert Boulter is an English film, television and theatre actor. He had a guest @-@ starring role on the television series The Bill in 2000. This was followed by a starring role in the play Herons written by Simon Stephens, which was performed in 2001 at the Royal Court Theatre. He had a guest role in the television series Judge John Deed in 2002. In 2004 Boulter landed a role as \" Craig \" in the episode \" Teddy's Story \" of the television series The Long Firm ; he starred alongside actors Mark Strong and Derek Jacobi. He was cast in the 2005 theatre productions of the Philip Ridley play Mercury Fur, which was performed at the Drum Theatre in Plymouth and the Menier Chocolate Factory in London. He was directed by John Tiffany and starred alongside Ben Whishaw, Shane Zaza, Harry Kent, Fraser Ayres, Sophie Stanton and Dominic Hall. \n\n In 2006, Boulter starred alongside Whishaw in the play Citizenship written by Mark Ravenhill. He appeared on a 2006 episode of the television series, Doctors, followed by a role in the 2007 theatre production of How to Curse directed by Josie Rourke. How to Curse was performed at Bush Theatre in the London Borough of Hammersmith and Fulham. Boulter starred in two films in 2008, Daylight Robbery by filmmaker Paris Leonti, and Donkey Punch directed by Olly Blackburn. In May 2008, Boulter made a guest appearance on a two @-@ part episode arc of the television series Waking the Dead, followed by an appearance on the television series Survivors in November 2008. He had a recurring role in ten episodes of the television series Casualty in 2010, as \" Kieron Fletcher \". Boulter starred in the 2011 film Mercenaries directed by Paris Leonti. \n\n In 2000 Boulter had a guest @-@ starring role on the television series The Bill ; he portrayed \" Scott Parry \" in the episode, \" In Safe Hands \". Boulter starred as \" Scott \" in the play Herons written by Simon Stephens, which was performed in 2001 at the Royal Court Theatre. A review of Boulter's performance in The Independent on Sunday described him as \" horribly menacing \" in the role, and he received critical reviews in The Herald, and Evening Standard. He appeared in the television series Judge John Deed in 2002 as \" Addem Armitage \" in the episode \" Political Expediency \", and had a role as a different character \" Toby Steele \" on The Bill. \n\n He had a recurring role in 2003 on two episodes of The Bill, as character \" Connor Price \". In 2004 Boulter landed a role as \" Craig \" in the episode \" Teddy's Story \" of the television series The Long Firm ; he starred alongside actors Mark Strong and Derek Jacobi. Boulter starred as \" Darren \", in the 2005 theatre productions of the Philip Ridley play Mercury Fur. | Robert Boulter is an English film, television and theatre actor. He had a guest @-@ starring role on the television series The Bill in 2000. This was followed by a starring role in the play Herons production from 2004 at the Drum Theatre, adapted from John Tiffany and Ben Whishaw play Tyger : a family lost in time at the Menier Chocolate Factory in 2005, won many critical accolades. In 2006, Boulter starred with Shane Zaza in a Constellations, a co - production directed by Harry Kent at the Bush Theatre, where the actors included Fraser Ayres, Sophie Stanton and Dominic Hall. This play was about dementia and was typical Boulter - light fare, but also showcased Boulter's fine facial features and his exceptional ability to stage a deeply moving and emotive role. After starring in another Whishaw production, Tyrant, in 2006, Boulter appeared again in the same production at Hammersmith's Bush Theatre, directed by Mark Ravenhill. Zaza became his colleague in the 2008 production of The Long Firm, which also starred Josie Rourke as the Prime Minister and Boulter as a scorpion that resembles the prime minister, and their take on the writing of The Sting brought Boulter further recognition. After another of Boulter's roles, 2002's Devil Doll, this time he was also playing Toto the dog, and played the title role in an adaptation of The Duellists at the Theatre Royal, Plymouth. Boulter continued to receive plaudits for his performances in the Devil Doll and Curse, two productions at London's Bush Theatre which directed by the theatre's artistic director, Jonathan Kent, starring children's entertainers such as Simon Stephens and Simon Callow. London's Bush Theatre then produced Boulter's production of the play for an international theatre festival, Romeo & Juliet, in May 2008. It ran at the London Lesbian, Gay & Bisexual festival and featured the titles Donkey Punch and Thank Heaven For Me. The April Production at the ExCeL Centre in November 2008 was a double bill written by Boulter and starring the actor Paris Leonti as the main lead and the undervalued resident theatre actor, Toby Steele. The production of Boulter's play Waking the Dead, to be produced by Paris Leonti at the Royal Court Theatre in 2010, is a co - production with Kamila Shamsie. In 2011 Boulter's play Sentences, with the writer Kieron Fletcher, opened on stage at the Drum Theatre in London with two plays from Boulter's personal collection. His production of The Independent on The Sugar Loaf, which was a top ten hit for the Menier Chocolate Factory Dominic Hall had a brief role in a Boulter play, The Long Firm, in 2000, directed by Fraser Ayres at The Lyric, Southampton in 2001, about mental health and dementia. |

Figure 12: Example of human-written and ENGIN-generated Wikipedia articles (from WikiText dataset (Merity et al., 2016)). We generate articles conditioned on both the title **Robert Boulter**, and the first 50 tokens highlighted by light yellow. The articles are cut to fit the figure size.

**1933 Treasure Coast hurricane**

Human-written Wikipedia

The 1933 Treasure Coast hurricane was the second @-@ most intense tropical cyclone to strike the United States during the active 1933 Atlantic hurricane season. The eleventh tropical storm, fifth hurricane, and the third major hurricane of the season, it formed east @-@ northeast of the Leeward Islands on August 31. The tropical storm moved rapidly west @-@ northwestward, steadily intensifying to a hurricane. It acquired peak winds of 140 miles per hour ( 225 km / h ) and passed over portions of the Bahamas on September 3, including Eleuthera and Harbour Island, causing severe damage to crops, buildings, and infrastructure. Winds over 100 mph ( 161 km / h ) affected many islands in its path, especially those that encountered its center, and many wharves were ruined. \n\n Subsequently, it weakened and made landfall at Jupiter, Florida, early on September 4 with winds of 125 mph ( 201 km / h ). The hurricane moved across the state, passing near Tampa before moving into Georgia and dissipating. In Florida, the strong winds of the cyclone blew buildings off their foundations, and numerous trees were prostrated in citrus groves. The Treasure Coast region received the most extensive destruction, and Stuart, Jupiter, and Fort Pierce were heavily damaged. Inland, the cyclone weakened rapidly but produced prodigious amounts of rain, causing a dam to collapse near Tampa. The storm caused $ 3 million in damage ( 1933 USD ) after damaging or destroying 6 @,@ 848 homes. \n\n Unusually, the storm hit Florida less than 24 hours before another major hurricane bearing 125 @-@ mile @-@ per @-@ hour ( 201 km / h ) winds struck South Texas ; never have two major cyclones hit the United States in such close succession. \n\n The origins of the hurricane were from a tropical wave that possibly spawned a tropical depression on August 27, although there was minimal data over the next few days as it tracked to the west @-@ northwest. On August 31, a nearby ship reported gale force winds, which indicated that a tropical storm had developed to the east @-@ northeast of the Lesser Antilles. Based on continuity, it is estimated the storm attained hurricane status later that day. Moving quickly to the west @-@ northwest, the storm passed north of the Lesser Antilles and Puerto Rico. Early on September 2, a ship called the Gulfwing reported a barometric pressure of 978 mbar ( 28 @.@ 88 inHg ), which confirmed that the storm attained hurricane status. After passing north of the Turks and Caicos islands, the hurricane struck Eleuthera and Harbour Island in the Bahamas on September 3, the latter at 1100 UTC. A station on the latter island reported a pressure of 27 @.@ 90 inHg ( 945 mb ) during the 30 minute passage of the eye.

ENGIN-generated Wikipedia

The 1933 Treasure Coast hurricane was the second @-@ most intense tropical cyclone to strike the United State during the active 1933 Atlantic hurricane season. The eleventh tropical storm, fifth hurricane. A Category 4 cyclone with winds of 100 mph generated winds of 161 km per hour and a water temperature of -49 degrees. The storm killed 6 people, flooded an area of 848 sq km, and caused $ 3 million in damage. Although it missed the Florida coast, the system swept across the rest of the United States less than 24 hours. The heat from the system caused widespread fires and continued to bring heavy rain and wind throughout the south and central regions. In south Florida alone, the fires killed two men and destroyed 98 homes in the Lesser Antilles and southwest Florida. The Turks and Caicos islands were particularly badly hit because of a storm surge that produced high seas and flooding, especially along the coast from Tampa, Georgia, to Puerto Rico. The Turks and Caicos islands experienced over 125 mph winds, winds of 125 @-@ mile per hour, and only 1 cm to 2 inches of rain in the 30 minute interval starting around 1400 UTC. The Turks and Caicos islands also experienced an over 140 mph wind shear at 5875 kp / m. In south Georgia, home to the Simpson and Penally counties, the worst of the damage was felt in the Savannah area, where 800 + homes were damaged or destroyed and well over 1000 had to be demolished. The most intense tropical storm to strike the United States during the 1933 Atlantic hurricane season occurred when Tropical Depression 6 formed over the Bahamas during the winter of 1913. The storm resulted in catastrophic loss of life and destruction. As of 24 km west of the Bahamas, the lowest pressure was 3.21 mb, and the storm had maximum sustained winds of 200 km/h, with gusts to over 200 km/h. In south Florida, the intense storm produced winds of 100 mph over south - western southwest Florida and a south - western coast generally between the edges of the Low Range and the extreme southeast of the storms. In western Cuba, the storms produced winds up to what the Weather Bureau described as \" about 65 mph \" in some spots, and all tropical storms at Category 4, 5 and 7, respectively, were downgraded to a tropical storm or depression. Texas though was hit by Hurricane Norbert in early October 1913. At the time, the Weather Bureau reported that the storm produced winds of 800 - 900 km/h, and a storm tide of 848 m. The storm passed over northeast Texas between midnight and dawn, and was the first major tropical cyclone to impact the United States during the winter of 1913. Norbert resulted in 848 lives being lost and $ 98 m in damages, amongst other catastrophic damage, as well as numerous fatalities.

Figure 13: Example of human-written and ENGIN-generated Wikipedia articles about **1993 Treasure Coast hurricane**.

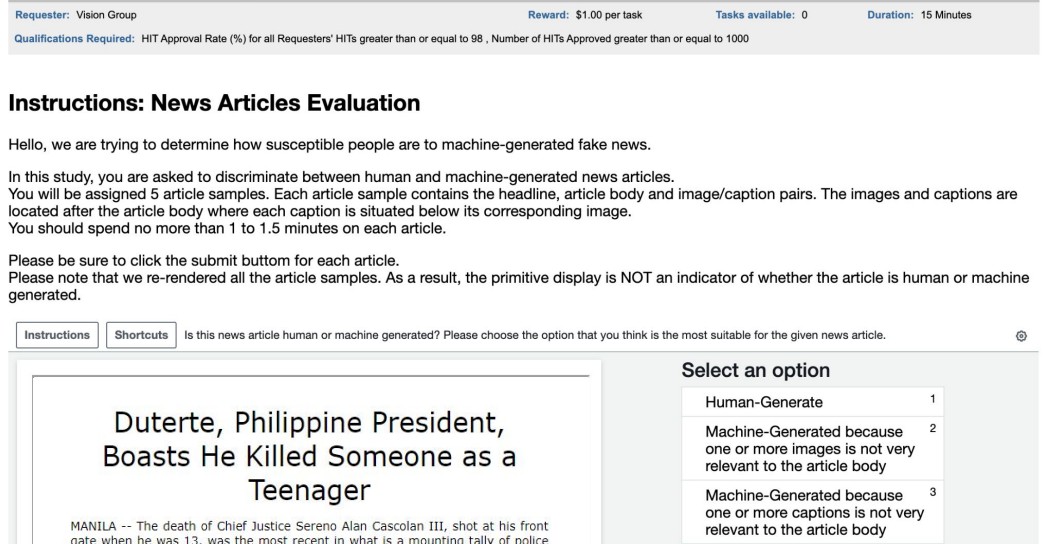

Figure 14: The interface used by AMT workers in our article quality annotation experiment.

