# OpenReview forum: "Show and Write: Entity-aware Article Generation with Image Information"
_ICLR.cc/2023/Conference — Submitted to ICLR 2023_

### Official Review · Reviewer_ewLW · 2022-10-26

**Confidence:** 5
**Correctness:** 3
**Technical Novelty And Significance:** 2
**Empirical Novelty And Significance:** 2
**Recommendation:** 3

**Clarity, Quality, Novelty And Reproducibility:**

The paper is very clear and easy to follow. The authors explain their method and appropriately cite the work it builds upon with sufficient detail to reproduce their results.
The paper is well put together and of high quality. However, the experiments and design of the method and model is a straightforward extension of existing techniques. Many obvious avenues for improving the model, such as improving the CLIP-based NER selection mechanism were not explored, so the method itself is of lower quality.
Overall, the method is of incremental novelty over comparable methods like GROVER. The model does not take in any visual features and the paper is not the first to condition news article generation on entities, as the InfoSurgeon paper did this (and actually conditioned on more complex relation types while integrating image-only visual information).

**Details Of Ethics Concerns:**

The proposed approach is a text generation approach comparable to GROVER.
GROVER has been used for generating misinformation online.
The experiments show that the articles generated by the authors' method are HARDER FOR  HUMANS TO DETECT as machine-generated.
The technoology could be used to spread disinformation that is harder to detect than existing methods.

**Strength And Weaknesses:**

[Strengths]
The idea of incorporating entity information into the generator is an important one. Existing methods for generating news articles, like GROVER, usually take in a title and generate the article based on that (and maybe the source it is supposed to be from). The article, however, fails to mention critical entities that would be mentioned in an article about that subject, making the generated article less realistic. By incorporating entity information into the generator, the model is able to generate more realistic articles.

Similarly, the authors attempt to integrate "image information". The authors argue that news image captions are different from typical image captioning in that the captions are much less visually grounded. The authors argue that directly integrating visual features is not idea and instead also condition their article generator on the captions. In this way, the generated article is much more likely to be related to the visual content paired with it.

Experimentally, the authors demonstrate that their method is better able to fool humans and the machine discriminator, suggesting that the articles it generates are more naturally. The authors also demonstrate impressive gains in perplexity in their experiments and the qualitative results bear out the quality of the generation.

[Weaknesses]
There are a number of weaknesses of the proposed approach.

The model is in some sense an extension to methods like GROVER. GROVER is quite similar to the proposed method in that GROVER trains a news article generator conditioned on metadata. In this paper, the authors essentially just re-train GROVER, but now add two additional metadata fields to the conditioning - one is the list of entities to be mentioned and the other is the *ground truth* image caption. From a technical perspectivie, there is little novelty to this approach. All the authors are doing is retraining GROVER using additional conditioning. Their is no novelty in the learning scheme, despite their being room for it.

For example, in the learning process used by ENGIN, no loss constraints enforce that ENGIN actually mentions all the named entities in the generated article. People have typically tackled this problem by using pointer-networks or similar strategies to perform a copying of named entities from the input into the output. The authors do not do this and it is unclear why not. Moreover, there is a lack of explicit loss to enforce named entities to appear and to penalize the model for their not appearing, save the standard text generation loss. Why don't authors enforce the model to actually use these entities and not ignore them?

Most importantly from my perspective is that critical related work is not cited. The authors are not the first to condition news article generation on entities or visual information. InfoSurgeon (Fung et al, ACL 2021, oral) presents a method for detecting machine generated news articles. As part of the paper, the authors train a news article generation method. The generator takes in a knowledge graph consisting of entities, relations between entities, events, as well as purely visual entities detected using image features as well as image events detected using imSitu. In sum, the input to the model and its reliance on visual features is much richer than ENGIN.

As it stands, ENGIN relies on ground truth image captions to generate the article and incorporate visual information. However, as it stands, the model is never able to mention facts that are purely observable visually - for example, maybe an image shows three people cheering but the caption and entity list don't mention this. ENGIN is unable to make any such claim in the article text because it doesn't ingest any visual features. I also am somewhat concerned at the way the paper claims to take in "Image Information" but doesn't actually make use of any purely visual features (objects, image events, etc.) and instead relies on captions.

The CLIP-based NER mechanism is very simple. The authors just use CLIP to take an image's embedding and retrieve a set of candidate entities. Given that entities are a key component of your model, I would have expected a more rigorous or trained model to improve the entity selection mechanism.

**Summary Of The Paper:**

The authors propose a method for generating news articles conditioned on entities to appear in an article. The authors claim that existing methods for generating news articles only model text and ignore visual content and secondly that these methods do not explicitly account for named entities to be mentioned in generated articles. The authors argue that this makes the machine generated articles less natural and prone to failing to mention entities that should be mentioned and which would make the article more realistic.
The authors propose a method, called Engin, for generating news articles conditioned on entity information. Engin takes as input metadata and a list of entities to appear in the article and is trained to generate news articles.
Engin also desires to take "image information" into account in generating the article. However, Engin is *not* conditioned on visual features from the image. Instead, the model uses the *ground truth" captions as the representation of the image.
For providing named entities to the model, the authors consider two techniques. One is providing the ground truth entities to appear (oracle) and another is using a CLIP retrieval technique to discover these entities automatically.

Authors experimentally evaluate their method at generating news articles conditioned on entities and "image information". They compare their method against a number of recent techniques (GPT2, Grover, etc.) and show improved perplexity on their generated articles on two datasets (VisualNews and GoodNews).
The authors also provide a number of supplemental experiments. They performed a user study to assess how well humans could detect the generated articles and found that their articles were harder to detect than the baseline articles (hence higher quality). They also performed other ablations, testing the impact of different metadata fields on generation, the ability for machine discriminators to detect their articles, and for zero-shot article generation.
The authors also provide qualitative examples of the articles generated by their method.

**Summary Of The Review:**

The authors present a method for generating news articles that mention entities. As explained, the paper is well put together and the appropriate experiments are performed. However, the method is a straightforward extension of existing work and offers little technical novelty - it is highly similar to GROVER just adding an additional conditioning metadata field essentially.
Obvious avenues for improvement, such as improving CLIP-based NER were not performed.
Similarly, relevant work (e.g. InfoSurgeon) was not compared against or cited.
Visual features are not used by the model.

At this time, I would argue the paper does not offer a sufficient contribution for publication as a conference paper.

---

> ### Author Response · Authors · 2022-11-19
> **Response to ewLW**
>
> We thank the reviewer for the comments, we appreciate their time and will use their suggestions to improve our paper.
>
> >The model is in some sense an extension to methods like GROVER. GROVER is quite similar to the proposed method in that GROVER trains a news article generator conditioned on metadata. In this paper, the authors essentially just re-train GROVER, but now add two additional metadata fields to the conditioning - one is the list of entities to be mentioned and the other is the ground truth image caption. From a technical perspectivie, there is little novelty to this approach. All the authors are doing is retraining GROVER using additional conditioning. Their is no novelty in the learning scheme, despite their being room for it.
>
> As we mentioned in Section 3.4, ENGIN and GROVER use the same architecture as GPT2. However, it does not mean that these three methods are the same. The training strategies of these three methods are different. GROVER uses metadata to replace title as the prompt compared to GPT2. ENGIN further introduces visual information and entity-aware mechanism on GPT2. We compare ENGIN with finetuned GPT2 in Table 1. If we just re-train GROVER, we will not see that ENGIN notably outperforms finetuned GPT2 and GROVER across all three model sizes on different datasets.
>
> >For example, in the learning process used by ENGIN, no loss constraints enforce that ENGIN actually mentions all the named entities in the generated article. People have typically tackled this problem by using pointer-networks or similar strategies to perform a copying of named entities from the input into the output. The authors do not do this and it is unclear why not. Moreover, there is a lack of explicit loss to enforce named entities to appear and to penalize the model for their not appearing, save the standard text generation loss. Why don't authors enforce the model to actually use these entities and not ignore them?
>
> We have updated our paper to include the comparison and discussion in Table 2.
> Specifically, using loss constraints to enforce the model mentions all named entities can not get better language representations. For example, DBA [1] lexically constrained the pre-specified words to appear in the output. Since the algorithm is applied to the decoding process, the model itself does not learn any improved language representations for downstream tasks. To further validate the improvement of our model, we trained BDA on GoodNews using the code provided by SOCKEYE [3]. From the Table, We see that the performance of BDA on GoodNews is much worse than our model (51.2 vs. 12.0). There are two major reasons that may drop the performance of BDA. First, since BDA mainly focuses on sequence-to-sequence translation, the model does not effectively leverage the prior knowledge from pretrained generative language models such as GPT2. Second, though these entities are enforced to appear in the output, the models may not put all named entities in correct positions.
>
> **Comparison to baselines with copy mechanism**
> |   Method   | GoodNews |
> |----------- |:--------:|
> | BDA [1] |  51.2  |
> | InfoSurgeon [2] | 41.8  |
> | ENGIN | 12.0 |
>
> [1] Post, Matt, and David Vilar. "Fast lexically constrained decoding with dynamic beam allocation for neural machine translation." arXiv preprint arXiv:1804.06609 (2018).
>
> [2] Fung, Yi, et al. "Infosurgeon: Cross-media fine-grained information consistency checking for fake news detection." Proceedings of the 59th Annual Meeting of the Association for Computational Linguistics and the 11th International Joint Conference on Natural Language Processing (Volume 1: Long Papers). 2021.
>
> [3] Hieber, Felix, et al. "Sockeye: A toolkit for neural machine translation." arXiv preprint arXiv:1712.05690 (2017).

---

> > ### Author Response · Authors · 2022-11-19
> > **Additional Response to ewLW**
> >
> > > Most importantly from my perspective is that critical related work is not cited. The authors are not the first to condition news article generation on entities or visual information. InfoSurgeon (Fung et al, ACL 2021, oral) presents a method for detecting machine generated news articles. As part of the paper, the authors train a news article generation method. The generator takes in a knowledge graph consisting of entities, relations between entities, events, as well as purely visual entities detected using image features as well as image events detected using imSitu. In sum, the input to the model and its reliance on visual features is much richer than ENGIN.
> >
> > We have updated our paper to include the comparison and discussion in Table 2 of our paper. We would like to point out that InfoSurgeon [2] uses a knowledge graph extracted from an existing article and then manipulates it to generate new text.  Thus, it cannot be used to generate a new article without significant manual input to create the knowledge graph.  However, to provide a comparison, we provide a knowledge graph as an additional input to DBA [1]. In Table 2, we observe the perplexity of InfoSurgeon is 41.8, which is much worse than ours (41.8 vs. 12.0). The reason is similar to BDA [1]. Since InfoSurgeon considers article manipulation as a sequence-to-sequence translation task, it may not effectively leverage the prior knowledge from pretrained generative language models.
> >
> > We note that at the time of this writing, we only have performed this comparison on GoodNews, and we will update our paper for our pending results using BDA and InfoSurgeon on VisualNews when they are available.
> >
> > >As it stands, ENGIN relies on ground truth image captions to generate the article and incorporate visual information. However, as it stands, the model is never able to mention facts that are purely observable visually - for example, maybe an image shows three people cheering but the caption and entity list don't mention this. ENGIN is unable to make any such claim in the article text because it doesn't ingest any visual features. I also am somewhat concerned at the way the paper claims to take in "Image Information" but doesn't actually make use of any purely visual features (objects, image events, etc.) and instead relies on captions.
> >
> > We included the method using purely visual features (bottom-up feature baseline (Anderson et al., 2018)) in Table 2(a), which performs much worse than our approach (26.6 PPL vs. 14.8 using CLIP-predicted named entities on GoodNews).  This poor performance is likely due to the loose correlation between image content and the articles (discussed in Section 3.2 in more detail).   Thus, we did not directly use image features for our final model as our experiments showed predicting named entities is far more effective.  We also note that it would be very challenging for any method to make predictions for objects it had never seen before.  In other words, even if we were to include image features, it would not represent any object that was not seen in its training data well.  Thus, using purely visual features may still find settings like those the reviewer suggests very challenging, which is partly reflected in our results.
> >
> >
> >
> > >The CLIP-based NER mechanism is very simple. The authors just use CLIP to take an image's embedding and retrieve a set of candidate entities. Given that entities are a key component of your model, I would have expected a more rigorous or trained model to improve the entity selection mechanism.
> >
> > The task of our paper is not developing a visual-based NER model. Since CLIP-based NE already notably improves the performance of our language model, we found it sufficient to demonstrate how such a model could be used in our approach. However, it does point out that our method can be further improved using a better image-text matching framework without additional effort, although demonstrating this would not change any of our conclusions.
> >
> > [1] Post, Matt, and David Vilar. "Fast lexically constrained decoding with dynamic beam allocation for neural machine translation." arXiv preprint arXiv:1804.06609 (2018).
> >
> > [2] Fung, Yi, et al. "Infosurgeon: Cross-media fine-grained information consistency checking for fake news detection." Proceedings of the 59th Annual Meeting of the Association for Computational Linguistics and the 11th International Joint Conference on Natural Language
> > Processing (Volume 1: Long Papers). 2021.
> >
> > [3] Hieber, Felix, et al. "Sockeye: A toolkit for neural machine translation." arXiv preprint arXiv:1712.05690 (2017).

---

### Official Review · Reviewer_5iPM · 2022-10-27

**Confidence:** 4
**Correctness:** 4
**Technical Novelty And Significance:** 3
**Empirical Novelty And Significance:** 2
**Recommendation:** 6

**Clarity, Quality, Novelty And Reproducibility:**

The paper is clearly written and easy to follow. The novelty is limited to the application of a well known concept/technique to a relatively new domain.

**Strength And Weaknesses:**

**Strengths**

-- The proposed method is quite elegant and simple and is easy to extend by following works.

-- The entity retrieval method (using entity retrieval from a pretrained image-text model) makes the approach more generalizable and not dependent on a provided list of entities.

-- The empirical results show effectiveness of the model as it outperforms all the baselines.

-- The paper is clearly written and easy to follow

**Weaknesses**

-- While the application of named entities as additional metadata is novel to this domain, the same has been studied extensively in NLP literature. It is a good observation that existing methods do not explicitly model named entities, but the technical novelty of the proposed solution is limited.

**Summary Of The Paper:**

This work is in the domain of text generation for long-form articles. The key contribution of this work is to provide additional metadata in the form of named entities to the generative model. Such entities are either provided in advance (oracle) or can also be derived using entity retrieval based on image associated with the article. The empirical results show the benefits of the approach.

**Summary Of The Review:**

Overall, while the method is quite simple and elegant and will propel more future work, this work has limited novelty in its current form.

---

> ### Author Response · Authors · 2022-11-19
> **Response to 5iPM**
>
> We thank the reviewer for the comments, we appreciate their time and will use their suggestions to improve our paper.
>
> > While the application of named entities as additional metadata is novel to this domain, the same has been studied extensively in NLP literature. It is a good observation that existing methods do not explicitly model named entities, but the technical novelty of the proposed solution is limited.
>
> The major difference between our work and most existing entity-aware methods (Liu et al., 2020, Dong et al., 2021) is that our method directly models the named entities, whereas those in prior work typically address this during post-processing.  While our experiments show that this is important for generating articles, it is also important for downstream applications.
>
> To illustrate, we performed an experiment on article-to-image retrieval task used in Tan et al., 2022 using the GoodNews dataset. We train a linear probe between features computing using ENGIN-base and Finetuned GPT2 from Table 1 of our paper and image features. Specifically, the image features are extracted by CLIP and language features are extracted from the last hidden layer of language models. Then, given an article, the goal is to retrieve the ground truth image from a set of 1K images.  We provide our results below
>
> **Article-to-image retrieval on GoodNews. We follow NewsStories (Tan et al., 2022) for evaluation**
> |   Method          | R@1 |  R@5 | R@10 |
> |-------------------------------|:----------:|-----------:|-----------:|
> | CLIP (Radford et al., 2021) | 17.14 | 33.77 | 43.56 |
> | MIL-NCE (Miech et al., 2020) | 15.50  | 28.96  | 37.60 |
> | Finetuned GPT2    |  16.08  |  31.99  |   41.45   |
> | ENGIN            | 18.24  |  35.62   |   45.36    |
>
> We see that ENGIN outperforms both the CLIP and MIL-NCE baselines used in Tan et al., 2022, as well as our own linear probe over finetuned GPT2.  This helps demonstrate the ability of our approach to generalize to other downstream tasks.  We updated our paper to include article-to-image retrieval experiments in Appendix A.7.

---

### Official Review · Reviewer_HgLz · 2022-10-27

**Confidence:** 3
**Correctness:** 3
**Technical Novelty And Significance:** 3
**Empirical Novelty And Significance:** 3
**Recommendation:** 6

**Clarity, Quality, Novelty And Reproducibility:**

This paper is clear and well-organized, I believe it is not difficult to reproduce the results.

**Strength And Weaknesses:**

This paper is well-written and easy to follow. The idea is simple but intuitive. Incorporating image information is very useful in article generation. The authors conducted extensive experiments and ablation tests to demonstrate the effectiveness of the proposed framework.

Weaknesses: the proposed model only extracted the entity information from the image, while images contain various information out of entities. I would suggest to add a discussion section to outlook how to fully use the image information in this task. In addition, the way of entity-aware article generation is simple (this is not a weakness), but other structures to restrict/prioritize identified entities during the generation should also be tried.

**Summary Of The Paper:**

This paper proposed an entity-award article generation framework that incoperates image information. By using external tools such SpaCy and CLIP, this framework can extract named entity candidates to supervise text generation.

**Summary Of The Review:**

Novel and interesting work, with good paper organization. But the structure of the framework is not fully explored.

---

> ### Author Response · Authors · 2022-11-19
> **Response to HgLz**
>
> We thank the reviewer for the comments, we appreciate their time and will use their suggestions to improve our paper.
>
> > The proposed model only extracted the entity information from the image, while images contain various information out of entities. I would suggest to add a discussion section to outlook how to fully use the image information in this task.
>
> We appreciate the suggestion by the reviewer.   We note that we provide this discussion in Section 4.2 of our paper. In particular, for a discussion on how to fully use the image information, we refer the reviewer to Table 2(a), where we report results using the bottom-up features from Anderson et al., 2018.  We find that using these image features directly performs much worse than our approach (e.g., it obtained 26.6 PPL on GoodNews compared to 14.8 using CLIP-extracted named entities).  We find this is likely due, in part, to the fact that articles are only loosely related to the images (discussed in Section 3.2 in more detail).
>
>
> > In addition, the way of entity-aware article generation is simple (this is not a weakness), but other structures to restrict/prioritize identified entities during the generation should also be tried.
>
> We thank the reviewer for the comment.  We would like to refer to you the results in Table 2(b), where we compare our entity-aware mechanisms like InjType and VNC that were proposed in prior work.  As can be seen, our approach obtains around 4.5 PPL improvement over these methods, helping to demonstrate their effectiveness.

---

### Official Review · Reviewer_2PvE · 2022-10-29

**Confidence:** 4
**Correctness:** 3
**Technical Novelty And Significance:** 2
**Empirical Novelty And Significance:** 2
**Recommendation:** 5

**Clarity, Quality, Novelty And Reproducibility:**

The method proposed in this paper is simple and therefore not difficult for the reader to understand. However, the details of the method and some statements still need further clarification. Although the application of image information in NLG is interesting, I think this paper is overclaimed for the utilization of image information. In essence, this paper is still a named entity conditioned generation and thus the novelty is limited.

**Strength And Weaknesses:**

+ Strength
  - The motivation of this paper is clear enough and it is interesting to introduce image and entity information in the article generation task. Image information augmented NLU and NLG has been explored in many works [1,2,3].
  -  The model proposed in this paper is simple and effective and achieves good results compared to the baselines.
+ Weaknesses
  - Although the authors claim that image information is introduced, the model actually makes use of ground-truth captions and entities extracted from the article. The image information is only used for CLIP to retrieve the relevant entities. Thus, in my opinion, the utilization of images in this paper is too shallow and the ENGIN is more like a named entity-conditioned article generation work. Also, how would it work if use textual information (metadata) instead of image information to retrieve related entities from the candidate entities?
  - There are some details of the method that I don't understand well enough. For example,
     1. In inference, the generated text may contain annotations of entity types, right? then what do you do with these entity type annotations?
     2. Is the utilization of named entities reflected in two aspects? The first is the generation conditioned on named entities. The second is the annotation of named entities and their categories in the target sequence during training.
     3. When using image information for named entity retrieval, do you use entity names directly or construct specific prompts? Can you further analyze and evaluate the performance of named entity retrieval based on oracle NEs.
  - I think the statements are inaccurate in many places, e.g.
    1. 'our key contribution is a novel Entity-aware mechanism to help our model recognize and predict the entity names in articles'. The purpose of this paper is not to recognize entities. This statement would be confusing.
    2. 'captions and named entities extracted from images'. I can't agree with the statement that the entities are extracted from the images, because the image information is only used to filter the entities recognized by Spacy.
    3. 'existing methods model named entities uniformly with the other text, making the leverage of named entities less effective.' Can you explain the reasons for this? Or provide some experimental results to support it.


[1] Visualize Before You Write: Imagination-Guided Open-Ended Text Generation

[2] Imagination-Augmented Natural Language Understanding

[3] Visually-Augmented Language Modeling

**Summary Of The Paper:**

This paper designs an entity-aware and image-information-integrated article generation method. The entities in the article are recognized with the help of image information and labeled with entity types. The paper claims that by annotating these entities on a target sequence, it enables the model to perceive entity information. The proposed model is evaluated on two news datasets and zere-shot Wikitxt and achieves excellent performance compared to the baselines.

**Summary Of The Review:**

Overall, I don't think the contribution of this paper is sufficient to be accepted as a conference paper.

---

> ### Author Response · Authors · 2022-11-19
> **Response to 2PvE**
>
> We thank the reviewer for the comments, we appreciate their time and will use their suggestions to improve our paper.
>
> >Although the authors claim that image information is introduced, the model actually makes use of ground-truth captions and entities extracted from the article. The image information is only used for CLIP to retrieve the relevant entities. Thus, in my opinion, the utilization of images in this paper is too shallow and the ENGIN is more like a named entity-conditioned article generation work.
>
> One of the primary contributions of this work is the study of how we can effectively use the image information to generate a news article.  In our experiments, we find that predicting named entities from images is more effective than directly using the images. For example, comparing the first column of Table 2(a), which uses the bottom-up features (Anderson et al., 2018) to directly use the images, we get only 26.6 PPL on GoodNews, whereas using CLIP predicted named entities gets 14.8 PPL, a gain of almost 12 points. In our paper (e.g., Section 4.2), we discussed that the reason could be the loose correlation between image content and their articles.  We find this no different than other vision-language models that use object/attribute predictors to help locate phrases [1] or to answer questions about an image [2].
>
> [1] Phrase Localization Without Paired Training Examples, ICCV, 2019.
>
> [2] Solving Visual Madlibs with Multiple Cues. BMVC, 2016.
>
>
> > Also, how would it work if use textual information (metadata) instead of image information to retrieve related entities from the candidate entities?
>
> We also presented the experiments using entities from textual information in Figure 4 (Cap-EA vs. Cap-CapNE-EA). We observe that using textual information instead of image information does not bring any additional improvement (17.1 vs. 17.1 on VisualNews, 15.9 vs. 16.0 on GoodNews), whereas using CLIP to extract named entities reduces PPL to 16.1 and 14.8 on VisualNews and GoodNews, respectively.
>
> > There are some details of the method that I don't understand well enough. For example,1. In inference, the generated text may contain annotations of entity types, right? then what do you do with these entity type annotations?
>
> Entity types are used to explicitly model named entities. We attach each named entity by its corresponding type in our language model. We use special tokens as the indicator of entity types so that the model can distinguish named entities from other text in documents. While we are happy to clarify this further, we would also refer the reader to Section 3.3 for additional details. We have updated our paper to clarify this.
>
> >Is the utilization of named entities reflected in two aspects? The first is the generation conditioned on named entities. The second is the annotation of named entities and their categories in the target sequence during training.
>
> Yes, the utilization of named entities is reflected in two aspects and we observe that both aspects can boost the performance. An ablation study of these two aspects is presented in Figure 4.
>
> > When using image information for named entity retrieval, do you use entity names directly or construct specific prompts? Can you further analyze and evaluate the performance of named entity retrieval based on oracle NEs.
>
> We construct the NER candidate list by extracting named entities from articles in the dataset. Then we use CLIP to select top-k entities given an image. We did evaluate the effect that named entities have in Figure 4 (Cap-CapNE-EA vs. Cap-ClipNE-EA vs. Cap-NE-EA). We observe that both ClipNE and NE improve the performance of our model, demonstrating the effectiveness of retrieved named entities.

---

> > ### Author Response · Authors · 2022-11-19
> > **Additional Response to 2PvE**
> >
> > > I think the statements are inaccurate in many places, e.g.'our key contribution is a novel Entity-aware mechanism to help our model recognize and predict the entity names in articles'. The purpose of this paper is not to recognize entities. This statement would be confusing.'captions and named entities extracted from images'.
> >
> > This statement points out the contribution of our work instead of task definition. Right before this sentence (the first few sentences in our abstract), we discussed the issue that named entities are difficult to be correctly recognized and predicted by prior work. Therefore, we purpose a novel method to better recognize and predict entity names in articles, improving performance. We have updated our paper to further clarify this.
> >
> > > I can't agree with the statement that the entities are extracted from the images, because the image information is only used to filter the entities recognized by Spacy.
> >
> > In our experiments, we use CLIP to extract named entities from a list, which is no different than any other classification task that would predefine a list of categories to predict.  This list of named entities could be extracted automatically from any data from a similar distribution as your target data using Spacy or any other named entity extractor.
> >
> >
> > > 'existing methods model named entities uniformly with the other text, making the leverage of named entities less effective.' Can you explain the reasons for this? Or provide some experimental results to support it.
> >
> >  In Figure 4, we did compare methods that explicitly model named entities (Cap-EA) with methods that model named entities uniformly (Cap). We observe that combining language models with our Entity-aware mechanism can get a minimum gain of 0.6 Perplexity. See Section 4.2 for discussion.

---

### Official Review · Reviewer_LeSY · 2022-10-30

**Confidence:** 4
**Clarity, Quality, Novelty And Reproducibility:** writing is good; technical originalit…
**Correctness:** 3
**Technical Novelty And Significance:** 2
**Empirical Novelty And Significance:** 3
**Recommendation:** 6

**Strength And Weaknesses:**

Strong & Weak:

S1: It is novel to exploit image info in the form of textual entities extracted from it. In this way, the info contained in image and article body is now in the same/homogeneous feature space to be input to the generation model. The side effect is to explicitly alleviate the heterogeneity across the original text space and visual space in the input layer instead of the hidden layers. This may make the learning of modality fusion easier. Hope the authors can discuss furthermore from this perspective.

W1: One possible weakness in my own opinion: While the proposed ENGIE is a new method for article generation with image recommender, each of the individual piece of ENGIE (entities extraction by SpaCy Python library, ranking candidate entities by CLIP, standard controllable text generation model) is not new in itself and has been widely adopted in existing works as pointed out in the paper. This may be not a big issue in practice, since ENGIE demonstrated a strong empirical result on three datasets.


**Summary Of The Paper:**

Summary:

This paper proposed to care about article generation with image information where the image info is used in an entity-aware way instead of directly using the visual features. That is, the embedded image in the associated article is firstly transformed into the textual space in the form of textual captions and textual named entities. After the image transformation, the textual contents extracted from the image act like normal body texts of an article, and are then integrated into standard controllable text generation models. To differentiate the entity texts from other texts, the entity categories are appended to the entity texts so the model can learn to predict the labels of named entities as well. Extensive experiments on three datasets are conducted with different model sizes.


**Summary Of The Review:**

Comments:

C1: The ENGIE model keeps the top $k=10$ candidate entities according to the similarity scores computed by the CLIP, any parameter cures varying with this $k$ to show its impact?

C2: Furthermore, it seems (not clearly said in the paper) that the similarity score is independently computed between each individual entity and the image, and does not consider the relationships among different entities. (This is motivated by the CRF decoding strategy in named entity recognition where the labels/categories of the entities are related in a sequence.) For example, the top $k$ entities, say E1,E2,…, have the maximum similarity scores independently, but can we find a semantic-dense (defined in a proper way) subgraph G’, say it contains E1’,E2’,…, which has larger sum similarity score SUM(G’{E1’,E2’,…}) than that of SUM(G{E1,E2,…}). Hope some discussions are given in this direction.

---

> ### Author Response · Authors · 2022-11-19
> **Response to LeSY**
>
> We thank the reviewer for the comments, we appreciate their time and will use their suggestions to improve our paper.
>
> >C1) The ENGIN model keeps the top candidate entities according to the similarity scores computed by the CLIP, any parameter cures varying with this to show its impact?
>
> We have updated our paper to include additional experiments on the number of detected entities in Appendix A.4. We see that our model achieves the best performance when k is set to 15. When k is less than 10, the improvement is limited.
>
> **Ablation study on the number of named entities detected by CLIP (GoodNews).**
> |   top-k entities   | 5 | 10 | 15 | 20 |
> |----------- |:--------:| :--------:| :--------:| :--------:|
> | PPL |  15.5  | 14.8 | 14.5 | 14.6 |
>
>
> > C2) Furthermore, it seems (not clearly said in the paper) that the similarity score is independently computed between each individual entity and the image, and does not consider the relationships among different entities. (This is motivated by the CRF decoding strategy in named entity recognition where the labels/categories of the entities are related in a sequence.) For example, the top entities, say E1,E2,…, have the maximum similarity scores independently, but can we find a semantic-dense (defined in a proper way) subgraph G’, say it contains E1’,E2’,…, which has larger sum similarity score SUM(G’{E1’,E2’,…}) than that of SUM(G{E1,E2,…}). Hope some discussions are given in this direction.
>
> We appreciate the suggestion proposed by the reviewer. The incorporation of relationships among different entities is definitely an improvement worth trying. We have updated our paper to include the discussion in Appendix B (Limitations and Future Work). Specifically, our current methods detect named entities from images considering each entity independently using a text-image matching framework. However, since the relationships between entities also affect the probability that entities appear in images, the incorporation of entity relationships can be considered to further improve the entity detection module.

---

### Official Review · Reviewer_g2Tq · 2022-11-03

**Confidence:** 3
**Correctness:** 3
**Technical Novelty And Significance:** 2
**Empirical Novelty And Significance:** 2
**Recommendation:** 5

**Clarity, Quality, Novelty And Reproducibility:**

The paper is generally well-written. I think the method lack novelty. The method seems straightforward and can be reproduced.


**Strength And Weaknesses:**

Strength
1. The writing is clear and the proposed method seems straightforward and easy to implement.

Weaknesses
1. I’m strongly skeptical about the prospect of using this technology to generate news articles, as the authors seem to aim at. Any words (especially nouns and verbs) hallucinated from the model instantly make the intended news article “fake news”. See the example in Figure 5. How much of the generated news article is true? Did Lizaso actually say that? This is also frightening as it sounds plausible (in a news article tone).

2. The method lacks novelty. The image information essentially is only used to extract name entities (the caption is provided).


**Summary Of The Paper:**

This paper proposes a method for article generation given visual information. Visual features from the embedded images are used to extract name-entity based on the CLIP model. Language model conditioned on the extracted entities along with other meta information is utilized to generate the body of the text. Experiments on three datasets including two news datasets show the efficacy of the model compared to GPT baselines.


**Summary Of The Review:**

I think the paper overclaims that it uses image information and the method is not novel enough for publication.

---

> ### Author Response · Authors · 2022-11-19
> **Response to g2Tq**
>
> We thank the reviewer for the comments, we appreciate their time and will use their suggestions to improve our paper.
>
> >I’m strongly skeptical about the prospect of using this technology to generate news articles, as the authors seem to aim at. Any words (especially nouns and verbs) hallucinated from the model instantly make the intended news article “fake news”. See the example in Figure 5. How much of the generated news article is true? Did Lizaso actually say that? This is also frightening as it sounds plausible (in a news article tone).
>
> Like many other models evaluated on their ability to generate text such as the GPT family of models, our model is not limited to improving the performance of this setting.  This is illustrated in prior work where good language models can provide benefits to many downstream tasks  (Radford et al., 2018; 2019; Brown et al., 2020).  For example, Tan et al., 2020 demonstrated that existing language generation models, including those like GROVER which was trained explicitly to generate news articles, often does not represent named entities effectively, which can be used to help detect automatically generated articles that may be used to spread misinformation.  However, these deficiencies are easily countered by making small edits to the generated text, which would make these automatically generated articles more challenging to detect.  Our approach could be used to simulate these settings, enabling us to generate more challenging data with which to train these detectors.
>
>
> That said, we expect our approach to be beneficial to a number of settings requiring reasoning about text that may refer to many named entities like news articles or Wikipedia entries.  For example, learning how to relate images and articles, which could be used to help quickly identify a good image to include with your article or to identify semantic mismatches between existing articles and images, which may also indicate manipulated content.  To illustrate this, we evaluate our ENGIN model on the GoodNews benchmark following the article-to-image retrieval benchmark from Tan et al., 2022.  We train a linear probe between features computing using ENGIN-base and Finetuned GPT2 from Table 1 of our paper and image features. Specifically, the image features are extracted by CLIP and language features are extracted from the last hidden layer of language models. Then, given an article, the goal is to retrieve the ground truth image from a set of 1K images.  We provide our results below
>
> **Article-to-image retrieval on GoodNews. We follow NewsStories (Tan et al., 2022) for evaluation**
> |   Method          | R@1 |  R@5 | R@10 |
> |-------------------------------|:----------:|-----------:|-----------:|
> | CLIP (Radford et al., 2021) | 17.14 | 33.77 | 43.56 |
> | MIL-NCE (Miech et al., 2020) | 15.50  | 28.96  | 37.60 |
> | Finetuned GPT2    |  16.08  |  31.99  |   41.45   |
> | ENGIN            | 18.24  |  35.62   |   45.36    |
>
> We see that ENGIN outperforms both the CLIP and MIL-NCE baselines used in Tan et al., 2022, as well as our own linear probe over finetuned GPT2.  This helps demonstrate the ability of our approach to generalize to other downstream tasks.  We updated our paper to include article-to-image retrieval experiments in Appendix A.7.
>
>
> >  The method lacks novelty. The image information essentially is only used to extract name entities (the caption is provided).
>
> We highlight two major differences between our method and prior work. First, we leverage named entities and captions as our image representations to improve the performance of article generation, which is unexplored by prior work. Second, we purpose an entity-aware mechanism to demonstrate that explicitly modeling entity names can improve performance even without image information.  Note that ours is not the first to propose an entity-aware mechanism, and we discuss the comparison to these prior methods in detail in the third paragraph of the Introduction.  We also perform experiments demonstrating that our entity-aware mechanism is more effective than those in prior work in Table 2(b).  Thus, we strongly disagree with the reviewer’s characterization of our work, and would be happy to discuss the differences between our work and any prior methods suggested by the reviewer.

---

### Author Response · Authors · 2022-12-08
**Further Discussion**

We thank the reviewers for their comments and suggestions. We would like to know if there are other questions or inquiries about our paper, and we are more than happy to address any additional comments. We appreciate your time and look forward to your update.

---

> ### Comment · Reviewer_ewLW · 2022-12-09
> **Ethical Issues**
>
> How do authors propose to address ethical issues?
> We see with Chat-GPT there are safeguards in place to prevent abuse.
> InfoSurgeon fake article generator was not released, but the authors released the detector.
>
> Have you thought about publicly releasing your detector but not generator?
> You might want to restrict generator to researchers with an agreement to prevent distribution?
> This might prevent abuse.

---

> > ### Author Response · Authors · 2022-12-11
> > **Response to Reviewer ewLW**
> >
> > >How do authors propose to address ethical issues?
> >
> > Like all AI models, ours does have the potential for misuse. Specifically, our model can be used to improve the generation of articles where there are key terms that require special care (e.g., Wikipedia, news, scientific articles). On one hand, our method enables us to better search and understand these articles, which could be used to help recommend more effective charts by learning the relationship between desired and perceived takeaways. On the other hand, it also can be useful in misinformation detection. For example, in DiDan (Tan et al. (2020), the authors found that mismatches in named entities between articles and image captions was an effective cue to train both human and automatic detectors to recognize misinformation. In this case, since articles generated by our model naturally have more consistent named entity information, using data generated by our approach should help a misinformation method better recognize human generated as well as slightly edited machine-generated misinformation.
> >
> > Moreover, we note that discussing and publishing our work could lead to more users taking advantage of these advances, for good or ill. We draw an analogy to work in computer security, where publishing work on vulnerabilities in current systems could lead to bad actors attempting to use these problems. However, not publishing this work does not mean that these issues disappear, the vulnerability is still there and must still be corrected.  We feel that publishing work like ours provides a similar benefit, where it highlights a need to address a potential vulnerability to reduce the potential for misinformation to spread, while also enabling beneficial applications.
> >
> > More specific discussions can be found under the response to reviewer g2Tq. In the camera ready, we will also expand our discussion to reflect more of the points we made here.
> >
> > >We see with Chat-GPT there are safeguards in place to prevent abuse. InfoSurgeon fake article generator was not released, but the authors released the detector.
> >
> > >Have you thought about publicly releasing your detector but not generator? You might want to restrict generator to researchers with an agreement to prevent distribution? This might prevent abuse.
> >
> > Thank you for the suggestion. We do intend to put safeguards and incensing agreements in place that are similar to those in related work (e.g., similar to [GROVER (Zellers et al., 2020)](https://docs.google.com/forms/d/1LMAUeUtHNPXO9koyAIlDpvyKsLSYlrBj3rYhC30a7Ak)).  We also note that InfoSurgeon's contributions are primarily as a detector, unlike ours where the contributions are in the generator (i.e., it makes more sense they would release a detector rather than a generator).
> >
> > We also note that while we believe the generations produced by our method would be useful in creating a more powerful detector, as noted earlier, we did not develop one ourselves.  The detector used in our experiments in Table 3 is the publicly available one from OpenAI, as noted in that table's caption, so it is already available and need not be released by us.

---

### Decision · Program_Chairs · 2023-01-20

**Decision:**

Reject

**Justification For Why Not Higher Score:**

Overall all reviewers felt that the paper is not suitable for publication due to limited technical departure of prior work, some assumptions on existence of ground truth captions and a use case that seems limited from an applicative point of view.

**Justification For Why Not Lower Score:**

N/A

**Metareview: Summary, Strengths And Weaknesses:**

The authors propose to use visual information to extract named entities and generate articles from the named entities.
strengths: simple method that allows to easily generate more relevant text based on an image leading to better text generation
weaknesses: relatively limited technical departure from prior work, unconvincing use case, use of ground truth captions in generation.